# BayesENDS: Bayesian Electrophysiological Neural Dynamical Systems for Alzheimer's Disease Diagnosis

## Abstract

Alzheimer's disease (AD) alters Electroencephalogram (EEG) through slowed oscillations and diminished neural drive, yet most AD-EEG pipelines are black-box classifiers, lacking a unifying mathematical account of how both neural activity and its interaction dynamics evolve over time. We introduce BayesENDS, a Bayesian electrophysiological neural dynamical system that explores the possibility of incorporating neuron spiking mechanisms into a Bayesian neural dynamical system. By introducing a differentiable leaky-integrate-and-fire (dLIF) prior, BayesENDS is capable of inferring population events and interaction dynamics directly from EEG—without spike or interaction annotations. The dLIF prior encodes membrane dynamics, rate/refractory constraints, and physiologically plausible frequency ranges, improving identifiability while yielding biologically plausible, subject-level biomarkers alongside AD predictions. Across synthetic event-sequence benchmarks and real AD EEG datasets, BayesENDS delivers superior performance to state-of-the-art baseline methods.

## 1 Introduction

Alzheimer's disease (AD) is a progressive neurodegenerative disorder with growing global impact. Electroencephalography (EEG) provides a non-invasive, low-cost window into brain function and consistently shows *oscillatory slowing* in AD—power increases in delta/theta and decreases in alpha/beta—together with alterations in large-scale interactions and synchrony (Jeong, 2004; Dauwels et al., 2010; Babiloni et al., 2021). While deep learning has advanced EEG-based AD assessment, most pipelines remain *black-box classifiers* (Ieracitano et al., 2020; Pineda et al., 2019; Vicchietti et al., 2023; Tawhid et al., 2025) optimized for accuracy from hand-crafted or learned features, offering limited insight into how *neural activity* and *interaction dynamics* co-evolve over time (Ehteshamzad et al., 2024; Acharya et al., 2025; Wang et al., 2024b; Klepl et al., 2024).

Two technical obstacles motivate a unifying, electrophysiology-aware dynamical framework. First, scalp EEG is a noisy, frequency-dependent *linear mixture* of mesoscopic sources; recovering latent neuron population activity is an ill-posed inverse problem sensitive to modeling choices (Michel & Brunet, 2019; Michel et al., 2004). Second, interaction metrics face *interpretational pitfalls* (volume conduction, common input, SNR differences) and can yield inconsistent estimates across analysis pipelines unless dynamics and biophysical constraints are handled explicitly (Bastos & Schoffelen, 2016; Mahjoory et al., 2017).

We address these gaps with **BayesENDS**, a *Bayesian electrophysiological neural dynamical system* that infers event-driven latent dynamics and a conditional interaction graph directly from multichannel EEG. Concretely, BayesENDS (a) represents per-channel activity with an **Event Posterior Differential Equation (EPDE)** whose solution yields expected next-event times; (b) samples inter-event intervals via a **Mean–Evolving Lognormal Process (MELP)**, where the EPDE outputs parameterize the means

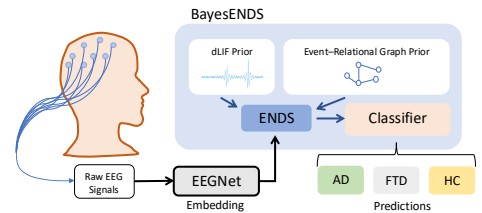

Figure 1: Overview of the BayesENDS pipeline

of a log-normal mixture with reparameterized
sampling; (c) imposes an electrophysiology-informed **differentiable leaky–integrate–and–fire (dLIF) prior** that encodes leak, refractory/rate constraints, and plausible frequency ranges; and (d) infers a directed **event–relational graph** (ERG) by mapping cross-channel event lags through a smooth nonlinearity into edge weights. The entire model is trained end-to-end under a variational inference framework; our analysis provides a tractable IVP-based bound for the event–prior KL under the dLIF rate and establishes ERG stability to lag noise.

**Major Contributions:**

- A unified Bayesian neural dynamical system that infers latent events and event–relational graph dynamics directly from EEG *without* spike or edge annotations.

- An electrophysiology-informed dLIF prior integrated into training, providing biophysical rate and refractory constraints.

- Theory establishing a computationally tractable IVP-based upper bound for the learning objective and a stability bound for BayesENDS's inferred graph dynamics.

- Empirical evidence showing: (i) accurate recovery of latent event and graph dynamics that improve understanding of AD; and (ii) superior performance over strong baselines on both synthetic benchmarks and real AD EEG datasets.

## 2 RELATED WORK

Most AD–EEG studies cast diagnosis as supervised classification over hand-crafted spectral/connectivity features or learned representations, achieving strong performance but offering limited mechanistic insight into how timing and interactions co-evolve. Recent Scientific Reports papers exemplify this trend: comprehensive pipelines comparing computational methods for AD classification, and multi-stage classification across the AD spectrum during memory-encoding versus rest (with higher accuracy during task-evoked states) (Vicchietti et al., 2023; Kim et al., 2024). Earlier machine-learning work integrates engineered EEG features (often spectral/topographic) within multimodal classifiers for dementia discrimination (Ieracitano et al., 2020). Recent large-scale and representation-learning approaches for AD–EEG continue this predominantly discriminative perspective: LEAD builds a foundation model for AD detection from multi-dataset EEG (Wang et al., 2025), COMET introduces hierarchical contrastive learning for medical time series including AD–EEG (Wang et al., 2023), Medformer employs multi-granularity Transformer patching for disease classification with AD cohorts (Wang et al., 2024a), and manifold-based vector-field modeling reconstructs high-density AD–EEG dynamics from low-density recordings (Peach et al., 2023). In contrast, our approach models *latent event dynamics* and a conditional interaction graph jointly, providing a generative account of how event timing and cross-channel lags give rise to predictive structure in EEG.

## 3 PROBLEM FORMULATION

We study unsupervised latent–event and relation discovery in multichannel time series with sequence-level labels. Given a dataset $\mathcal{D} = \{(X^{(n)}, Y^{(n)})\}_{n=1}^{N}$ with $X^{(n)} = \{x_c^{(n)}\}_{c=1}^{C}$ (e.g., multichannel EEG) and labels $Y^{(n)} \in \mathcal{Y}$ (e.g., AD vs. control), and *no* supervision on per-channel events or inter-channel relations, the objective is to infer (i) channel-wise latent event dynamics and (ii) a (possibly time-varying) relational/graphical structure among channels. For each channel $c \in \{1, \ldots, C\}$ and sequence $n$, let $T_c^{(n)} = \{t_{c,k}^{(n)}\}_{k=1}^{K_c^{(n)}}$ denote the (unknown) latent event times, where $t_{c,k}^{(n)}$ is the $k$-th event time on channel $c$ in sequence $n$, and $K_c^{(n)}$ is the (latent) number of events on that channel. Let $\mathbf{p}^{(n)}(t) = \{p_c^{(n)}(t)\}_{c=1}^{C}$ denote the corresponding posterior event-time distributions. The relational structure is represented as a graph process $G^{(n)}(t)$ with adjacency $A^{(n)}(t) \in \mathbb{R}^{C \times C}$. We aim to recover $\{\mathbf{p}^{(n)}(t)\}_{n=1}^{N}$ and the conditional graph dynamics $P\big(G^{(n)}(\cdot) \mid \mathbf{p}^{(n)}(\cdot)\big)$ from $\{X^{(n)}\}_{n=1}^{N}$, while using $\big(\mathbf{p}^{(n)}(\cdot), G^{(n)}(\cdot)\big)$ as inputs to a downstream predictor for $Y^{(n)}$; importantly, $Y^{(n)}$ does not supervise the latent events or relations directly.

The interaction strength between channels is modeled as a function of the temporal *co-occurrence and ordering* of inferred events. This assumption is grounded in neurobiological mechanisms of *spike-timing-dependent plasticity* (STDP), where near-coincident pre- and post-synaptic spikes modulate synaptic efficacy, forming the basis for capturing dynamic interactions between different neural regions(Bi & Poo, 1998; Feldman, 2012).

# 4  BAYESIAN NEURAL DYNAMICAL SYSTEM

## 4.1  OVERVIEW

We introduce BayesENDS, a Bayesian neural dynamical system for multichannel sequences that represents each channel with a latent *event* process and couples channels through a conditional *event–relational graph* (ERG) driven by the timing and ordering of inferred events. This design is motivated by settings where the clinically relevant signal resides in *when* events occur and *how* they align across channels. In EEG for Alzheimer's disease, for instance, oscillatory slowing and disrupted coordination suggest that event timing and cross-channel alignment are predictive, while neurobiological plasticity links near-coincident spikes to stronger coupling.

At a high level, BayesENDS consists of three interacting components. First, an *event posterior differential equation* (EPDE) summarizes per-channel event dynamics by producing posterior distributions over latent event times $\{T_c^{(n)}\}$ given the observed multichannel sequence $X^{(n)}$. Second, a *mean–evolving lognormal mechanism* (MELP) uses EPDE outputs as mean parameters to generate stochastic inter-event timing between successive latent event times $t_{c,k}^{(n)}$, ensuring positive and flexible (potentially multimodal) timing statistics. Third, an *event–relational graph* $G^{(n)}(t)$ is inferred from event co-occurrence and cross-channel lags derived from $T^{(n)}$, via an STDP-shaped mapping that encodes how the timing of events on one channel modulates effective coupling to others.

For each labeled sequence $(X^{(n)}, Y^{(n)})$, the triple $(X^{(n)}, T^{(n)}, G^{(n)})$ is passed to a decoder $p_\theta\big(Y^{(n)} \mid X^{(n)}, T^{(n)}, G^{(n)}\big)$ for downstream prediction (e.g., AD vs. control). Training is end-to-end via variational learning that jointly optimizes the EPDE and MELP while learning ERG dynamics under weak regularization; further details are given in the learning subsection.

## 4.2  LEARNING

We train BayesENDS end-to-end through variational inference by *minimizing* the negative evidence lower bound (ELBO). For labeled data $\{(X^{(n)}, Y^{(n)})\}_{n=1}^N$, let $T^{(n)}$ collect all latent event times $\{T_c^{(n)}\}_{c=1}^C$ and let $\tau^{(n)}$ collect the corresponding inter-event intervals $\{\tau_{c,k}^{(n)}\}$. The EPDE induces an approximate posterior $q_\phi\big(T^{(n)} \mid X^{(n)}\big)$, the MELP defines $q_\phi\big(\tau^{(n)} \mid X^{(n)}\big)$, and the decoder is $p_\theta\big(Y^{(n)} \mid X^{(n)}, T^{(n)}, G^{(n)}\big)$, where $G^{(n)}$ is the ERG associated with $X^{(n)}$ and $\eta$ denotes ERG parameters.

We write the ELBO as

$$\mathcal{L}_{\text{ELBO}}(\theta, \phi, \eta) = \sum_{n=1}^N \mathbb{E}_{q_\phi}\Big[ \log p_\theta\big(Y^{(n)} \mid X^{(n)}, T^{(n)}, G^{(n)}\big)\Big] \tag{1}$$

$$- \text{KL}_T^{(n)} - \text{KL}_\tau^{(n)} \tag{2}$$

$$+ \beta \, \mathcal{R}_{\text{ERG}}^{(n)} + \lambda_{\text{LIF}} \, \mathcal{R}_{\text{LIF}}^{(n)}, \tag{3}$$

where $\text{KL}_T^{(n)} := \text{KL}\big(q_\phi(T^{(n)} \mid X^{(n)}) \,\|\, p_{\text{dLIF}}(T)\big)$ compares the EPDE-induced path law to the electrophysiology-informed event prior $p_{\text{dLIF}}(T)$, and $\text{KL}_\tau^{(n)} := \text{KL}\big(q_\phi(\tau^{(n)} \mid X^{(n)}) \,\|\, p_0(\tau)\big)$ penalizes deviation from a lognormal(-mixture) prior over inter-event intervals. The term $\mathcal{R}_{\text{LIF}}^{(n)}$ softly enforces leaky–integrate–and–fire consistency on differentiable rate proxies read out from the EPDE state, with weight $\lambda_{\text{LIF}} \geq 0$. The term $\mathcal{R}_{\text{ERG}}^{(n)}$ is a weak, observable-based regularizer that nudges ERG edges toward experimental statistics computed from $X^{(n)}$ (e.g., correlation-based summaries), with strength $\beta \geq 0$.

**Challenges.** Three technical issues arise in optimizing equation 3. First, $\mathrm{KL}_T^{(n)}$ involves *path measures* induced by a differential equation and is intractable in closed form (it integrates over an infinite-dimensional trajectory); we therefore replace it with a tractable integral–rate surrogate that depends only on the dLIF rate $r(t)$, with a formal bound given in the Theory subsection. Second, enforcing a LIF prior directly is difficult because the spike function in LIF is *non-differentiable*, which prevents straightforward use in gradient-based training; instead we introduce differentiable rate proxies and constrain them to follow dLIF laws via $\mathcal{R}_{\mathrm{LIF}}^{(n)}$. Third, ERG learning lacks ground-truth edges; to avoid over-constraining the graph, we only use the weak regularizer $\mathcal{R}_{\mathrm{ERG}}^{(n)}$ to bias edge strengths toward experimental observables, leaving the fine-grained graph structure to be driven by event lags inferred from the EPDE–MELP posterior.

### 4.3 PRIOR: ELECTROPHYSIOLOGY–INFORMED dLIF PRIOR

We place a biophysical prior on latent event timing by instantiating each channel's latent events $T_c^{(n)} = \{t_{c,k}^{(n)}\}$ as a renewal process whose hazard is derived from a differentiable leaky–integrate–and–fire (dLIF) abstraction (Burkitt, 2006). For channel $c$, the (rescaled) membrane potential evolves as

$$\frac{d}{dt}u_c(t) = b_c(t) - u_c(t), \qquad b_c(t) > 1, \tag{4}$$

where $b_c(t)$ is an effective (learned) input drive. Given this membrane dynamics, the implied instantaneous *firing rate* is

$$r_c(t) = \Big[ -\log\big(1 - 1/b_c(t)\big)\Big]^{-1}. \tag{5}$$

This rate induces a dLIF inter-event time density

$$p_{\mathrm{dLIF},c}(t) = r_c(t)\,\exp\Big(-\int_0^t r_c(s)\,ds\Big), \tag{6}$$

and the resulting dLIF prior for channel $c$ is the renewal law $p_{\mathrm{dLIF}}(T_c^{(n)})$ with hazard $r_c(t)$. We parameterize $b_c(t)$ by a bounded neural mapping from learned embeddings, for example

$$b_c(t) = 1 + \mathrm{softplus}\big(g_\xi(z_c^{(n)}(t))\big), \tag{7}$$

where $z_c^{(n)}(t)$ denotes features derived from $X^{(n)}$. This construction guarantees $b_c(t) > 1$ and thus $r_c(t) > 0$. Absolute and refractory effects are incorporated through a smooth gating factor $\alpha_c^{(n)}(t) \in (0,1]$ constructed from recent events in $T_c^{(n)}$, using $r_c(t) = \alpha_c^{(n)}(t)\,r_c(t)$ to suppress implausible near–back–to–back spikes.

Because the hard spike nonlinearity is non–differentiable, we regularize *rates* rather than spikes. Concretely, the learning objective includes a dLIF consistency term

$$\mathcal{R}_{\mathrm{LIF}} = \sum_c \int_0^S \big(\widehat{r}_c^{(n)}(t) - r_c(t)\big)^2 dt, \tag{8}$$

where $\widehat{r}_c^{(n)}(t)$ is a differentiable rate proxy read from the EPDE state for sequence $n$ over a time horizon $[0, S]$. This encourages the learned rates to follow dLIF membrane dynamics without invoking non–differentiable spike functions (Neftci et al., 2019). The variational KL between the EPDE–induced path law $q_\phi\big(T_c^{(n)} \mid X^{(n)}\big)$ and $p_{\mathrm{dLIF}}(T_c^{(n)})$ is intractable in general; in the Theory subsection we replace it by a tractable integral–rate bound that depends on $r_c(t)$, yielding a stable surrogate for training while preserving the biophysical semantics of the prior.

### 4.4 POSTERIOR: EVENT POSTERIOR DIFFERENTIAL EQUATION (EPDE)

For each sequence $(X^{(n)}, Y^{(n)})$ and channel $c$, let $q_c^{(n)}\big(t \mid x_c^{(n)}\big)$ denote the density of the next event time given the observed channel signal $x_c^{(n)}$. If $\tilde{t}_{c,k-1}^{(n)}$ denotes the previous (predicted) event time on that channel, the expected next event time is

$$\tilde{t}_{c,k}^{(n)} = \int_{\tilde{t}_{c,k-1}^{(n)}}^{\infty} t\, q_c^{(n)}\big(t \mid x_c^{(n)}\big)\, dt. \tag{9}$$

To express this update via an initial value problem (IVP), we introduce an auxiliary function $\Phi_c^{(n)}(t)$ whose derivative accumulates the contribution of $q_c^{(n)}$:

$$\left(\Phi_c^{(n)}\right)'(t) \;=\; -\,t\,q_c^{(n)}\!\left(t \mid x_c^{(n)}\right). \tag{10}$$

Its initial value encodes the full expectation under $q_c^{(n)}$:

$$\Phi_c^{(n)}(0) \;=\; \int_0^\infty t\,q_c^{(n)}\!\left(t \mid x_c^{(n)}\right) dt. \tag{11}$$

With this definition, the expected next event time in equation 9 can be written as the IVP solution evaluated at the previous event time:

$$\tilde{t}_{c,k}^{(n)} \;=\; \Phi_c^{(n)}\!\left(\tilde{t}_{c,k-1}^{(n)}\right). \tag{12}$$

Directly solving equation 10–12 and computing $\Phi_c^{(n)}(0)$ is intractable, so we approximate this mapping with a differentiable neural surrogate that updates the predicted next event time:

$$\tilde{t}_{c,k}^{(n)} \;=\; f_{\theta_\Phi}\!\left(\tilde{t}_{c,k-1}^{(n)},\, x_c^{(n)}\right), \tag{13}$$

implemented to ensure $\tilde{t}_{c,k}^{(n)} > \tilde{t}_{c,k-1}^{(n)}$, so that latent events remain strictly ordered in time.

Consequently, differentiating the ideal $\Phi_c^{(n)}(t)$ with respect to $t$ yields an *event-time posterior* of the form

$$q_c^{(n)}\!\left(t \mid x_c^{(n)}\right) \;=\; -\,\frac{\left(\Phi_c^{(n)}\right)'(t)}{t}, \tag{14}$$

which we approximate with the EPDE parameterization. Across channels, the family $\{q_c^{(n)}(t \mid x_c^{(n)})\}_{c=1}^C$ provides a parametric approximation to the event-time posteriors $\{p_c^{(n)}(t)\}_{c=1}^C$ introduced in the problem formulation, and jointly defines the EPDE-induced posterior $q_\phi\!\left(T^{(n)} \mid X^{(n)}\right)$ over latent event times $T^{(n)} = \{T_c^{(n)}\}_{c=1}^C$ used in the variational objective. The predicted next-event times $\tilde{t}_{c,k}^{(n)}$ then serve as mean parameters for the mean–evolving lognormal mechanism (MELP) that models stochastic variability in event timing, as detailed in the next subsection.

## 4.5 SAMPLING: MEAN–EVOLVING LOGNORMAL PROCESS (MELP)

For each sequence $X^{(n)}$ and channel $c$, given the previous event time $\tilde{t}_{c,i-1}^{(n)}$, the EPDE produces a $K$-dimensional vector of *candidate mean intervals* $\tilde{\boldsymbol{\tau}}_c^{(n)} = \left(\tilde{\tau}_{c,1}^{(n)}, \dots, \tilde{\tau}_{c,K}^{(n)}\right) \in \mathbb{R}_+^K$, together with mixture weights $\mathbf{w}_c^{(n)} = \left(w_{c,1}^{(n)}, \dots, w_{c,K}^{(n)}\right) \in \Delta^{K-1}$ and scales $\mathbf{s}_c^{(n)} = \left(s_{c,1}^{(n)}, \dots, s_{c,K}^{(n)}\right) \in \mathbb{R}_+^K$. MELP draws the inter–event interval $\tau_{c,i}^{(n)}$ from a lognormal mixture:

$$p\!\left(\tau_{c,i}^{(n)} \mid \tilde{t}_{c,i-1}^{(n)}, X^{(n)}\right) \;=\; \sum_{j=1}^K w_{c,j}^{(n)}\, \mathrm{LogN}\!\left(\tau_{c,i}^{(n)};\, \mu_{c,j}^{(n)},\, (s_{c,j}^{(n)})^2\right), \tag{15}$$

where $\mathrm{LogN}(\cdot; \mu, s^2)$ denotes a lognormal density with log-mean $\mu$ and log-variance $s^2$.

For each component $j$, we choose $\mu_{c,j}^{(n)}$ as a function of $\tilde{\tau}_{c,j}^{(n)}$ and $s_{c,j}^{(n)}$ so that the *mean* of the corresponding lognormal distribution matches the EPDE-predicted interval $\tilde{\tau}_{c,j}^{(n)}$. This ties the mixture components' average inter-event times directly to the EPDE outputs, while the scales $s_{c,j}^{(n)}$ control uncertainty around these means.

Sampling from MELP is reparameterized to keep gradients pathwise:

$$k \sim \mathrm{Cat}\!\left(\mathbf{w}_c^{(n)}\right), \qquad \varepsilon \sim \mathcal{N}(0,1), \tag{16}$$

$$\tau_{c,i}^{(n)} = \exp\!\left(\mu_{c,k}^{(n)} + s_{c,k}^{(n)}\,\varepsilon\right), \qquad t_{c,i}^{(n)} = t_{c,i-1}^{(n)} + \tau_{c,i}^{(n)}. \tag{17}$$

During training we use a differentiable variant of equation 16 (e.g., Gumbel–Softmax) and take hard samples at test time. MELP guarantees positive inter-event intervals, captures multimodal timing statistics, and yields closed-form component-wise KL terms against a lognormal(-mixture) prior in the learning objective. Moreover, the mixture expectation $\mathbb{E}[\tau_{c,i}^{(n)}] = \sum_{j=1}^K w_{c,j}^{(n)} \tilde{\tau}_{c,j}^{(n)}$ is available in closed form and is used in downstream computations such as computing ERG lags.

Table 1: Toy dataset results by frequency band. CS: Cosine Similarity, IoU: Intersection-over-Union.

| Frequency Band (Hz) | Model | CS | Median Rate | 95% CI | IoU |
|---|---|---|---|---|---|
| [5, 10] | NODE | 0.951 | 1.000 | [1.000, 1.000] | 0.000 |
| [5, 10] | ODE-RNN | 0.951 | 1.000 | [1.000, 1.000] | 0.000 |
| [5, 10] | STRODE | 0.967 | 0.340 | [0.269, 0.410] | 0.000 |
| **[5, 10]** | BayesENDS (Ours) | **0.983** | **7.532** | **[4.300, 14.867]** | **0.473** |
| [10, 15] | NODE | 0.951 | 1.000 | [1.000, 1.000] | 0.000 |
| [10, 15] | ODE-RNN | 0.951 | 1.000 | [1.000, 1.000] | 0.000 |
| [10, 15] | STRODE | 0.964 | 0.251 | [0.153, 0.348] | 0.000 |
| **[10, 15]** | BayesENDS (Ours) | **0.982** | **12.503** | **[7.587, 24.918]** | **0.289** |
| [15, 20] | NODE | 0.951 | 1.000 | [1.000, 1.000] | 0.000 |
| [15, 20] | ODE-RNN | 0.951 | 1.000 | [1.000, 1.000] | 0.000 |
| [15, 20] | STRODE | 0.961 | 0.532 | [0.369, 0.695] | 0.000 |
| **[15, 20]** | BayesENDS (Ours) | **0.976** | **18.843** | **[10.465, 35.244]** | **0.202** |

## 4.6 EVENT–RELATIONAL GRAPH (ERG)

We weakly bias $\bar{A}^{(n)}$ toward observable statistics computed from $X^{(n)}$ (e.g., Pearson correlations $s_{ij}^{(n)}$ between channels $i$ and $j$) via a simple Fisher–$z$ alignment. We map both the observed and ERG-implied correlations into $z$-space:

$$z_{ij}^{\mathrm{obs},(n)} = \mathrm{atanh}\big(s_{ij}^{(n)}\big), \qquad z_{ij}^{\mathrm{pred},(n)} = \mathrm{atanh}\big(2\bar{A}_{ij}^{(n)} - 1\big), \tag{18}$$

and define the ERG regularizer as

$$\mathcal{R}_{\mathrm{ERG}}^{(n)} = \sum_{i<j} \left[ \frac{\big(z_{ij}^{\mathrm{obs},(n)} - z_{ij}^{\mathrm{pred},(n)}\big)^2}{2\,\sigma^2} + \frac{1}{2}\log\sigma^2 \right], \tag{19}$$

where $\sigma > 0$ is a (fixed or globally learned) scale parameter controlling the strength of the alignment. This regularizer (weighted by $\beta$ in the learning objective) encourages consistency between ERG-implied connectivity and experimental statistics, while still allowing the detailed edge structure to be driven primarily by event lags inferred from the EPDE–MELP posterior. The implementation details and additional theoretical results are provided in Appendix G and B, respectively.

## 5 EXPERIMENTS

In our experimental evaluations, we rigorously assessed the performance of BayesENDS across synthetic benchmarks and real Alzheimer's disease (AD) EEG datasets. First, we validated BayesENDS using synthetic event-sequence data, demonstrating its effectiveness in accurately inferring latent event dynamics compared to baseline neural ODE models. Subsequently, we conducted comprehensive experiments on two diverse EEG datasets (AD cohorts A and B), covering Alzheimer's disease, frontotemporal dementia, mild cognitive impairment, and healthy controls.

### 5.1 TOY DATASET

The toy dataset experiments evaluated how well BayesENDS and baseline neural ODE methods (NODE (Chen et al., 2018), ODE-RNN (Habiba & Pearlmutter, 2020), and STRODE (Huang et al., 2021)) recover latent event dynamics across distinct frequency bands ([5–10], [10–15], and [15–20] Hz). As summarized in Table 1, baseline methods achieved strong cosine similarity (CS) scores across all frequency bands, reflecting good sequence-level prediction performance. However, their intersection-over-union (IoU) scores were uniformly zero, indicating a fundamental limitation in capturing the latent event structure. This outcome highlights the common issue with purely data-driven neural approaches: despite excellent predictive accuracy, they often fail to recover the underlying generative mechanisms of data.

In contrast, BayesENDS maintained similarly high CS scores while notably achieving meaningful IoU values (e.g., 0.473 at [5–10] Hz, 0.289 at [10–15] Hz, and 0.202 at [15–20] Hz). These non-zero IoU scores demonstrate that BayesENDS successfully captures latent structures consistent with

Table 2: Results on Alzheimer's EEG datasets.

| Model | Dataset AD cohort A | | Dataset AD cohort B | |
|---|---|---|---|---|
| | Accuracy (%) | F1 (%) | Accuracy (%) | F1 (%) |
| EEGNet | 68.10 | 66.49 | 71.37 | 60.85 |
| LCADNet | 70.52 | 68.12 | 72.44 | 49.38 |
| LSTM | 70.52 | 68.24 | 77.89 | 61.35 |
| ATCNet | 64.71 | 60.98 | 71.09 | 50.92 |
| ADFormer | 69.35 | 65.28 | 82.38 | 63.89 |
| LEAD | 72.68 | 69.98 | 80.00 | 62.21 |
| **BayesENDS** | **75.03** | **72.69** | **89.82** | **64.87** |

Table 3: Ablation of spike-informed and connectome priors in **BayesENDS** on Alzheimer's EEG Datasets. Accuracy and F1 (%) reported as mean $\pm$ s.d. across runs.

| Dataset | Variant | Accuracy (%) | F1 (%) |
|---|---|---|---|
| **AD cohort A** | No prior | $70.52 \pm 11.83$ | $65.46 \pm 13.10$ |
| | dLIF prior | $73.92 \pm 9.84$ | $71.41 \pm 10.72$ |
| | ERG prior | $72.75 \pm 6.63$ | $70.35 \pm 7.91$ |
| | Dual priors | $\mathbf{75.03} \pm 8.29$ | $\mathbf{72.69} \pm 8.16$ |
| **AD cohort B** | No prior | $83.22 \pm 15.10$ | $60.72 \pm 12.37$ |
| | dLIF prior | $87.98 \pm 8.09$ | $62.95 \pm 9.24$ |
| | ERG prior | $86.20 \pm 9.96$ | $\mathbf{65.63} \pm 9.17$ |
| | Dual priors | $\mathbf{89.82} \pm 8.39$ | $64.87 \pm 10.67$ |

the generative process, particularly emphasizing the method's ability to infer interpretable and biologically plausible event dynamics. Moreover, the predicted event rates from BayesENDS showed wider but informative uncertainty intervals, aligning closely with the true frequency band intervals. Such uncertainty quantification highlights BayesENDS' capacity to provide both accurate and interpretable latent dynamics in noisy and ambiguous settings. Additional data generation protocol are provided in Appendix F.

## 5.2 ALZHEIMER'S DISEASE EEG DATASET

We extensively evaluated BayesENDS on two diverse EEG datasets covering Alzheimer's disease (AD), frontotemporal dementia (FTD), mild cognitive impairment (MCI), and healthy controls, as shown in Tabel 2. Across both datasets, BayesENDS outperformed state-of-the-art baselines such as CNNs, RNNs, and transformers. In Dataset AD cohort A, BayesENDS showed a clear ability to distinguish AD, FTD, and healthy participants, with consistently lower performance variability across runs. This stability highlights its effectiveness in capturing subtle neural dynamics despite EEG heterogeneity. In the more unbalanced Dataset AD cohort B, where distinctions between moderate AD, MCI, and healthy controls are subtler, BayesENDS still achieved the highest diagnostic accuracy. While some baselines showed significant performance fluctuations, BayesENDS remained robust, balancing sensitivity and specificity—demonstrating resilience to noise and distribution shifts common in real-world EEG data. Overall, BayesENDS effectively extracts clinically meaningful biomarkers from EEG, confirming its practical potential for accurate, interpretable AD diagnostics. Additional dataset descriptions, baseline methods, and experimental setup are presented in Appendix F.

## 5.3 ABLATION STUDIES

We ablated BayesENDS to assess each prior's impact on accuracy and interpretability (Table 3). On AD Cohort A, the no-prior baseline is moderate; adding the dLIF prior boosts accuracy, while the ERG prior improves accuracy, and F1 via inter-channel modeling. Combining both yields the best scores. On AD Cohort B, the pattern holds with larger gains: the dLIF prior gives the biggest accuracy lift under subtler classes, and the ERG prior raises F1. Their combination again performs best. Overall, each prior helps, and together they provide robust predictions on realistic EEG.

# 6 VISUALIZATIONS

Our visualizations emphasized BayesENDS' explainability. Specifically, we showcased the inferred frequency distributions from the dLIF prior across critical EEG channels, clearly linking decreased oscillatory frequency to increased disease severity. Additionally, the inferred event-relational graphs revealed meaningful connectivity patterns that align with known neurophysiological changes in dementia. Lastly, boundary time prediction comparisons highlighted BayesENDS' precision in temporal modeling compared to baseline methods, underscoring its superior capability to capture accurate event timings.

## 6.1 dLIF INFERRED FREQUENCY VISUALIZATION

To further explore and validate the interpretability of the latent neural dynamics captured by BayesENDS, we visualized the inferred frequency distributions derived from the differentiable leaky-integrate-and-fire (dLIF) prior across several key EEG channels associated with Alzheimer's disease progression, including channels F3, O2, Pz, and T3. Figure 2a exemplifies the frequency distributions at channel F3 for the Alzheimer's disease (AD), mild cognitive impairment (MCI), and healthy control (HC) groups. The distributions clearly illustrate a trend of decreasing frequency with increasing disease severity. Specifically, healthy controls exhibit the highest central frequency, indicating typical neural oscillatory activity. Subjects with MCI show slightly reduced frequency values, signifying the onset of neural slowing, while the AD group displays the lowest central frequencies, reflecting significant neural slowing commonly observed in Alzheimer's pathology. This consistent pattern across multiple critical EEG channels underscores the physiological relevance of BayesENDS' latent dynamics. The clear association between disease severity and decreased oscillatory frequency validates the biological interpretability of our model, highlighting its potential utility for understanding Alzheimer's disease progression and supporting clinical decision-making.

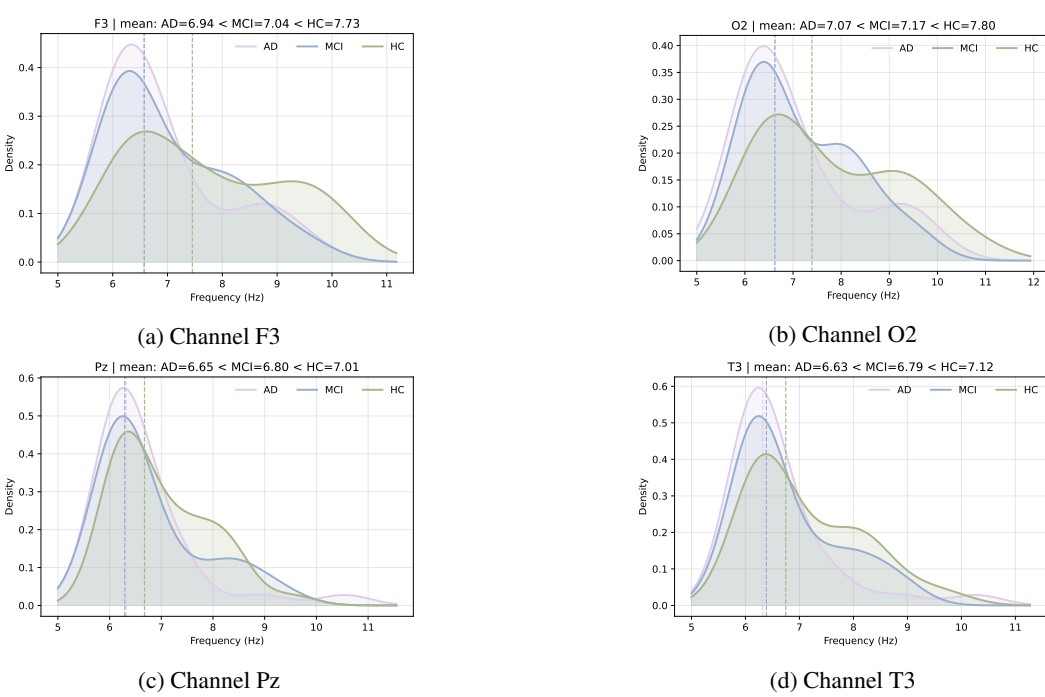

(a) Channel F3  (b) Channel O2

(c) Channel Pz  (d) Channel T3

Figure 2: Kernel density estimates of the inferred dLIF frequency distributions across Alzheimer's disease (AD), mild cognitive impairment (MCI), and healthy control (HC) groups for EEG channels F3, O2, Pz, and T3. The clear trend of decreasing central frequency with increasing disease severity illustrates the physiological relevance and interpretability of the BayesENDS model's inferred latent neural dynamics.

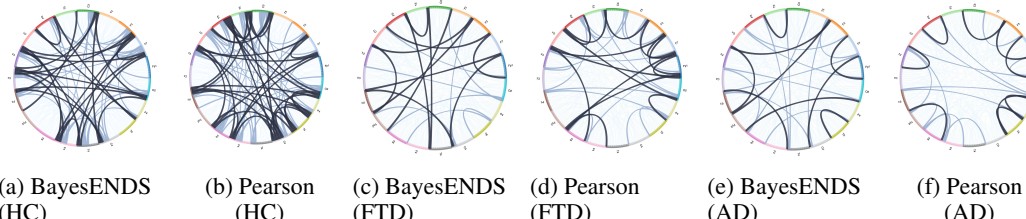

(a) BayesENDS (HC)  (b) Pearson (HC)  (c) BayesENDS (FTD)  (d) Pearson (FTD)  (e) BayesENDS (AD)  (f) Pearson (AD)

Figure 3: Comparison of EEG connectivity graphs inferred by BayesENDS versus Pearson correlation-based priors across healthy controls (HC), frontotemporal dementia (FTD), and Alzheimer's disease (AD) groups.

## 6.2 GRAPH CONNECTIVITY VISUALIZATION

We also visualized BayesENDS' inferred event-relational graphs (ERGs) against Pearson correlation-based connectivity graphs (Figure 3). Chord diagrams revealed distinct connectivity patterns across healthy controls (HC), FTD, and AD groups. BayesENDS' ERGs captured biologically plausible trends: HC showed dense, robust connectivity, while FTD and AD exhibited progressively sparser and weaker links. These patterns aligned with Pearson-derived graphs, validating the ERG's effectiveness. The coherence underscores ERG's ability to reflect dementia-related network degradation, strengthening model interpretability and clinical relevance.

## 6.3 BOUNDARY TIME PREDICTION VISUALIZATION

Figure 4 visually compares the predicted versus ground-truth boundary times for STRODE and BayesENDS across three distinct frequency bands ([5–10 Hz], [10–15 Hz], and [15–20 Hz]). STRODE demonstrates noticeable deviations from the ideal diagonal alignment, suggesting challenges in accurately recovering true boundary timings, particularly at higher frequency bands. In contrast, BayesENDS consistently maintains a tighter diagonal alignment across all frequency ranges, indicating superior accuracy and robustness in capturing underlying temporal structures. This visualization clearly illustrates BayesENDS' effectiveness in accurately inferring latent event boundaries in the toy dataset, reinforcing its suitability for precise temporal modeling in EEG data.

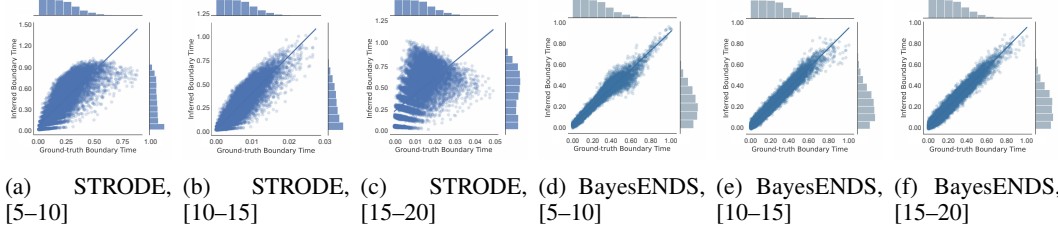

(a) STRODE, [5–10]  (b) STRODE, [10–15]  (c) STRODE, [15–20]  (d) BayesENDS, [5–10]  (e) BayesENDS, [10–15]  (f) BayesENDS, [15–20]

Figure 4: Predicted vs. ground-truth boundary times across frequency bands ([5–10], [10–15], [15–20] Hz): STRODE vs. BayesENDS.

## 7 CONCLUSION

We presented **BayesENDS**, a Bayesian electrophysiological neural dynamical system that infers latent event dynamics and a conditional event–relational graph directly from multichannel EEG. By coupling an Event Posterior Differential Equation (EPDE) with a Mean–Evolving Lognormal Process (MELP) and an electrophysiology-informed dLIF prior, the model yields identifiable, physiology-aware latents and supports end-to-end prediction. Our theory provides a tractable IVP-based upper bound for the event–prior KL and establishes stability of the inferred graph to lag noise, while experiments on synthetic and AD EEG data demonstrate superior accuracy and interpretable biomarkers.

## 8 ETHICS STATEMENT

Compliant with ICLR ethics. No human/animal subjects or IRB approval required. Public datasets used under license; no re-identification attempted. For academic use; real-world applications need further validation. No harmful insights, conflicts, or sponsorship. Possible dataset biases noted. Methods documented for reproducibility.

## 9 REPRODUCIBILITY STATEMENT

We have taken deliberate steps to facilitate the reproducibility of our results. A complete description of the model architectures, training procedures, and evaluation protocols is provided in the main text. All datasets used are publicly available and cited with their licenses.

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

## A LLM USAGE STATEMENT

In accordance with the ICLR 2026 submission guidelines, we disclose that large language models (LLMs) were used only for language editing and proofreading of this manuscript. No LLMs were employed for generating research ideas, designing methodologies, producing experimental results, or creating data. All scientific content, analysis, and conclusions were developed and verified by the authors. The authors take full responsibility for the integrity, originality, and accuracy of the submission.

## B THEORY

**Lemma 1** (Shift–stability of IVPs (Huang et al., 2021)). *Let $e > 0$ and $U \subset \mathbb{R}^n$ be open. Let $f_1, f_2 : [a - 2e, a) \to \mathbb{R}^n$ be continuously differentiable with $\|f_1'\| \leq M$ for some $M > 0$. Consider*

$$y_1'(t) = f_1(t), \quad y_1(a - e) = x_1, \qquad y_2'(t) = f_2(t) = f_1(t - e), \quad y_2(a - e) = x_2.$$

*Then, as $e \to 0^+$,*

$$\lim_{e \to 0^+} \left( \lim_{t \nearrow a} \|y_1(t) - y_2(t)\| \right) \leq \lim_{e \to 0^+} \|x_1 - x_2\|.$$

**Theorem 2** (IVP–based upper bound for the event–prior KL under dLIF rates). *Let $q(t)$ be a strictly positive, integrable density on $[0, S]$ ($0 < S \leq \infty$). Let the electrophysiology–informed prior be*

$$p_r(t) = r(t) \exp\left( -\int_0^t r(u)\, du \right),$$

*where $r : [0, S] \to [a, b] \subset (0, \infty)$ is measurable with $0 < a \leq b < \infty$. Define the change of variables $m = -e^{-t} \in [-e^{-S}, -1)$ and $M = -\log(-m) = t$. Set*

$$g(m) := -\frac{q(M)}{m\, M} \log \frac{q(M)}{M\, r(M)\, e^{-\int_0^M r(u)\, du}}, \qquad G'(m) = g(m), \quad G(-e^{-S}) = 0. \tag{20}$$

*Then*

$$\mathrm{KL}\big(q \,\|\, p_r\big) = \int_0^S q(t) \log \frac{q(t)}{r(t) e^{-\int_0^t r}}\, dt = \lim_{\varepsilon \downarrow 0} G(-\varepsilon), \tag{21}$$

*and for any $\varepsilon \in (0, e^{-S})$,*

$$\mathrm{KL}\big(q \,\|\, p_r\big) \leq G(-\varepsilon) + \big| G(-2\varepsilon) - G(-\varepsilon) \big| =: \mathcal{U}_\varepsilon, \tag{22}$$

*with $\mathcal{U}_\varepsilon \to \mathrm{KL}(q\|p_r)$ as $\varepsilon \downarrow 0$.*

In training we evaluate the KL term using the computable bound $\mathcal{U}_\varepsilon$ from Eq. 22 (fixed small $\varepsilon$ and an ODE solver for Eq. (20)).

We then analyze how entry–wise perturbations of lags affect the learned ERG when the edge map is exponential. For channels $i \neq j$ and time $t$, let the noise–free lag be $\Delta t_{ij}(t; T)$ and the perturbed lag be $\widetilde{\Delta t}_{ij}(t; T) = \Delta t_{ij}(t; T) + \xi_{ij}(t; T)$. Define the edge map $\phi_\alpha(x) = \exp(-\alpha|x|) \in [0, 1]$ with slope parameter $\alpha > 0$ and the (noise–free and perturbed) edges

$$e_{ij}(t; T) = \phi_\alpha(\Delta t_{ij}(t; T)), \qquad \widetilde{e}_{ij}(t; T) = \phi_\alpha(\widetilde{\Delta t}_{ij}(t; T)).$$

The decoder uses the Monte–Carlo, time–averaged adjacencies

$$\bar{A}_{ij} = \frac{1}{MS} \sum_{m=1}^{M} \sum_{s=1}^{S} e_{ij}(t_m; T^{(s)}), \qquad \widetilde{A}_{ij} = \frac{1}{MS} \sum_{m=1}^{M} \sum_{s=1}^{S} \widetilde{e}_{ij}(t_m; T^{(s)}).$$

**Theorem 3** (Entry–wise and matrix stability). *The exponential edge map is globally $\alpha$–Lipschitz: for all $x, y \in \mathbb{R}$,*

$$\left|\phi_\alpha(x) - \phi_\alpha(y)\right| \leq \alpha |x - y|. \tag{23}$$

*Consequently, for any $(i, j, t, T)$,*

$$\left|\widetilde{e}_{ij}(t; T) - e_{ij}(t; T)\right| \leq \alpha |\xi_{ij}(t; T)|. \tag{24}$$

*Averaging over time and Monte–Carlo samples yields the entry–wise bound*

$$\left|\widetilde{A}_{ij} - \bar{A}_{ij}\right| \leq \alpha \overline{|\xi_{ij}|}, \qquad \overline{|\xi_{ij}|} := \frac{1}{MS} \sum_{m=1}^{M} \sum_{s=1}^{S} |\xi_{ij}(t_m; T^{(s)})|, \tag{25}$$

*and the matrix (Frobenius–norm) bound*

$$\|\widetilde{A} - \bar{A}\|_{\mathrm{F}} \leq \frac{\alpha}{MS} \sum_{m=1}^{M} \sum_{s=1}^{S} \|\Xi^{(m,s)}\|_{\mathrm{F}}, \qquad \Xi^{(m,s)} := \left[\xi_{ij}(t_m; T^{(s)})\right]_{i \neq j}, \tag{26}$$

*hence $\|\widetilde{A} - \bar{A}\|_{\mathrm{F}} \leq \alpha \overline{\|\Xi\|_{\mathrm{F}}}$ with $\overline{\|\Xi\|_{\mathrm{F}}}$ the average Frobenius norm of lag–noise matrices.*

**Corollary 4** (Deterministic and probabilistic perturbation bounds). *(i) (Uniformly bounded noise). If $|\xi_{ij}(t; T)| \leq \varepsilon_\infty$ almost surely, then*

$$\left|\widetilde{A}_{ij} - \bar{A}_{ij}\right| \leq \alpha \varepsilon_\infty, \qquad \|\widetilde{A} - \bar{A}\|_{\mathrm{F}} \leq \alpha \varepsilon_\infty \sqrt{C(C - 1)}. \tag{27}$$

*(ii) (Sub–Gaussian noise). Suppose $\{\xi_{ij}(t_m; T^{(s)})\}_{m,s}$ are i.i.d., mean–zero, sub–Gaussian with proxy $\sigma^2$ (i.e., $\mathbb{E}e^{\lambda\xi} \leq \exp(\lambda^2\sigma^2/2)$). Then each difference $\Delta_{ij} := \widetilde{A}_{ij} - \bar{A}_{ij}$ is sub–Gaussian with proxy $\alpha^2\sigma^2/(MS)$ and*

$$\mathbb{P}(|\Delta_{ij}| \geq \tau) \leq 2 \exp\left(-\frac{MS\,\tau^2}{2\alpha^2\sigma^2}\right). \tag{28}$$

*If, in addition, $\xi \sim \mathcal{N}(0, \sigma^2)$, then*

$$\mathbb{E}\left[|\Delta_{ij}|\right] \leq \alpha\,\sigma\sqrt{\frac{2}{\pi}}, \qquad \mathbb{E}\left[\|\widetilde{A} - \bar{A}\|_{\mathrm{F}}\right] \leq \alpha\,\sigma\sqrt{\frac{2}{\pi}}\,\sqrt{C(C - 1)}. \tag{29}$$

**Implication.** Small perturbations in EPDE/MELP lags translate linearly (in $\alpha$) to entry–wise changes in the ERG, and averaging over samples/time further contracts the perturbation. Thus the ERG is provably stable to modest timing noise, with explicit constants controlled by the edge map slope $\alpha$ and the noise magnitude.

## C  PROOF OF THEOREM 2

*Proof.* By definition,

$$\mathrm{KL}(q\|p_r) = \int_0^S q(t) \log \frac{q(t)}{r(t)e^{-\int_0^t r}}\, dt.$$

Using $m = -e^{-t}$, we have $t = -\log(-m) = M$ and $dt = -\frac{1}{m} dm$. Hence

$$\mathrm{KL}(q\|p_r) = \int_{-e^{-S}}^{0} \frac{q(M)}{M} \log \frac{q(M)}{Mr(M)e^{-\int_0^M r}} \left(-\frac{1}{m}\right) dm = \int_{-e^{-S}}^{0} g(m)\, dm.$$

This improper integral equals $\lim_{\varepsilon\downarrow 0} \int_{-e^{-S}}^{-\varepsilon} g(m)\, dm = \lim_{\varepsilon\downarrow 0} G(-\varepsilon)$, proving Eq. (21). Since $r(t) \in [a,b]$ and $q$ is integrable, $g$ is locally integrable near $m = 0^-$. Split the integral at $-\varepsilon$ and at $-2\varepsilon$:

$$\int_{-e^{-S}}^{0} g = \int_{-e^{-S}}^{-\varepsilon} g + \int_{-\varepsilon}^{0} g \le G(-\varepsilon) + \left|\int_{-\varepsilon}^{0} g\right| \le G(-\varepsilon) + \left|\int_{-2\varepsilon}^{-\varepsilon} g\right| = G(-\varepsilon) + |G(-2\varepsilon) - G(-\varepsilon)|,$$

which yields Eq. (22). The last inequality uses Lemma 1 (applied to the IVPs $G_1'(m) = g(m)$ and $G_2'(m) = g(m)$ shifted by $\varepsilon$) to control the tail near the singular endpoint and the fact that $r$ is bounded in $[a,b]$, ensuring $g$ remains controlled as $m \to 0^-$. As $\varepsilon \downarrow 0$, the tail vanishes by dominated convergence, hence $\mathcal{U}_\varepsilon \to \mathrm{KL}(q\|p_r)$. $\qquad\square$

# D    PROOF OF THEOREM 3

*Proof. (Lipschitzness).* The absolute value is 1–Lipschitz: $||x| - |y|| \le |x - y|$. The function $u \mapsto e^{-\alpha u}$ on $u \ge 0$ has derivative $|-\alpha e^{-\alpha u}| \le \alpha$, hence it is $\alpha$–Lipschitz on $\mathbb{R}_{\ge 0}$. By composition of Lipschitz maps,

$$|\phi_\alpha(x) - \phi_\alpha(y)| = \left|e^{-\alpha|x|} - e^{-\alpha|y|}\right| \le \alpha \left||x| - |y|\right| \le \alpha |x - y|,$$

establishing Eq. (23). Taking $y = x + \xi$ gives Eq. (24).

*(Averaging).* Using linearity of the average and triangle inequality,

$$\left|\widetilde{\bar{A}}_{ij} - \bar{A}_{ij}\right| = \left|\frac{1}{MS} \sum_{m,s} \left(\widetilde{e}_{ij}(t_m; T^{(s)}) - e_{ij}(t_m; T^{(s)})\right)\right| \le \frac{1}{MS} \sum_{m,s} |\widetilde{e}_{ij} - e_{ij}| \le \frac{\alpha}{MS} \sum_{m,s} |\xi_{ij}(t_m; T^{(s)})|,$$

which is Eq. (25).

*(Matrix bound).* Define $\Delta^{(m,s)} := [\widetilde{e}_{ij}(t_m; T^{(s)}) - e_{ij}(t_m; T^{(s)})]_{i \ne j}$, so $\widetilde{\bar{A}} - \bar{A} = \frac{1}{MS} \sum_{m,s} \Delta^{(m,s)}$. By the triangle inequality of the Frobenius norm,

$$\|\widetilde{\bar{A}} - \bar{A}\|_{\mathrm{F}} \le \frac{1}{MS} \sum_{m,s} \|\Delta^{(m,s)}\|_{\mathrm{F}}.$$

Entry–wise inequality Eq. 24 implies $\|\Delta^{(m,s)}\|_{\mathrm{F}} \le \alpha \|\Xi^{(m,s)}\|_{\mathrm{F}}$, giving Eq. (26). $\qquad\square$

# E    PROOF OF COROLLARY 4

*Proof.* (i) From Eq. (25), $\overline{|\xi_{ij}|} \le \varepsilon_\infty$, giving the entry–wise claim. For the matrix bound, $\|\Xi^{(m,s)}\|_{\mathrm{F}} \le \varepsilon_\infty \sqrt{C(C-1)}$ for every $(m, s)$; apply Eq. (26).

(ii) For each fixed $(i, j)$, define i.i.d. variables $Y_{m,s} := \widetilde{e}_{ij}(t_m; T^{(s)}) - e_{ij}(t_m; T^{(s)})$. By Eq. (24), $Y_{m,s}$ is an $\alpha$–Lipschitz transform of $\xi_{ij}(t_m; T^{(s)})$, hence $Y_{m,s}$ is sub–Gaussian with proxy $\alpha^2\sigma^2$ (standard Lipschitz preservation of the $\psi_2$–norm). Then $\Delta_{ij} = (MS)^{-1} \sum_{m,s} Y_{m,s}$ is sub–Gaussian with proxy $\alpha^2\sigma^2/(MS)$, yielding Eq. (28) via the Chernoff bound.

For the Gaussian–mean bound, use $|Y_{m,s}| \le \alpha |\xi_{ij}(t_m; T^{(s)})|$ and linearity:

$$\mathbb{E}|\Delta_{ij}| \le \frac{\alpha}{MS} \sum_{m,s} \mathbb{E}|\xi| = \alpha \mathbb{E}|\xi| = \alpha \sigma \sqrt{\frac{2}{\pi}},$$

where the last equality is the mean absolute value of a zero–mean Gaussian. Summing the entry–wise bounds in quadrature gives the Frobenius expectation in Eq. (29). $\qquad\square$

# F ADDITIONAL EXPERIMENTAL DETAILS

## F.1 ALZHEIMER'S DISEASE EEG BASELINE METHODS

To rigorously evaluate our proposed method, we benchmarked it against several representative deep learning approaches commonly utilized for EEG analysis. These baselines include convolutional, recurrent, attention-based, and transformer-based models, each demonstrating distinct strengths for capturing various aspects of EEG signal patterns.

**EEGNet** (Lawhern et al., 2018) is a compact convolutional neural network initially developed for EEG-based brain–computer interfaces. It integrates depthwise and separable convolutions to effectively capture temporal, spatial, and frequency-specific characteristics in EEG data, making it a well-recognized lightweight yet powerful model in EEG classification tasks.

**LCADNet** (Kachare et al., 2024) is specifically tailored for Alzheimer's disease detection from EEG data. Utilizing optimized convolutional structures designed for computational efficiency without sacrificing discriminative power, LCADNet achieves competitive performance in resource-limited environments, making it a strong baseline for EEG-based AD diagnosis.

**LSTM** (Zhang & Yao, 2021) embodies recurrent neural networks tailored for modeling temporal dependencies inherent in EEG signals. By maintaining and updating hidden states across sequences, LSTMs effectively capture long-term dynamics and temporal correlations, making them naturally suitable for sequential EEG analyses.

**ATCNet** (Altaheri et al., 2022) employs a physics-informed architecture combining temporal convolutions with attention mechanisms. Originally proposed for motor imagery EEG classification, it effectively captures both local temporal details and global dependencies, showcasing adaptability across various EEG applications.

**ADformer** (Wang et al., 2024b) is a multi-granularity transformer specifically crafted for Alzheimer's disease evaluation using EEG signals. It utilizes multi-scale attention mechanisms to concurrently model fine-grained and coarse-grained temporal information, setting a high-performance standard in EEG-based AD diagnostics.

**LEAD** (Wang et al., 2025) exemplifies the recent advancement toward large-scale foundation models in EEG analysis. Pre-trained extensively on vast EEG datasets and fine-tuned for Alzheimer's disease detection, LEAD leverages transfer learning to provide robust, generalizable EEG representations, establishing a new benchmark in EEG-based clinical assessments.

## F.2 DATASET DESCRIPTIONS

### F.2.1 TOY DATASET AND DATA GENERATION

**Frequency bands and sampling.** To systematically evaluate our model's ability to capture latent event dynamics, we constructed synthetic datasets with clearly defined frequency bands. We generated latent event rates $\lambda$ from truncated normal distributions centered at the midpoint of each target frequency band: low band [5–10 Hz] with $\lambda \sim \text{TruncNormal}(\mu = 7.5, \sigma = 1.0; [5, 10])$, middle band [10–15 Hz] with $\lambda \sim \text{TruncNormal}(\mu = 12.5, \sigma = 1.0; [10, 15])$, and high band [15–20 Hz] with $\lambda \sim \text{TruncNormal}(\mu = 17.5, \sigma = 1.0; [15, 20])$. This design ensures concentrated event-rate distributions within each band while avoiding frequencies outside the desired range.

**Data scale and splitting strategy.** For each frequency band, we independently generated three data splits: a training set with 150 distinct event rates, each having 50 sequences (7,500 sequences total); a validation set with 25 new event rates and 50 sequences per rate (1,250 sequences total); and a test set with an additional 25 new event rates, again with 50 sequences per rate (1,250 sequences total). Importantly, no overlap of event rates occurs across training, validation, and test splits to ensure proper evaluation of model generalization.

**Sequence generation.** Each synthetic sequence comprises 20 observations, constructed by sampling inter-event times $\Delta t_i$ from an exponential distribution with parameter $\lambda$. The event timestamps

$t_i$ are obtained cumulatively by $t_i = \sum_{j=1}^{i} \Delta t_j$. Observations $y_i$ are subsequently generated using the relationship:

$$y_i = \sin(t_i) + \eta_i, \quad \eta_i \sim \mathcal{N}(0, \sigma_\eta^2),$$

where the default noise level is $\sigma_\eta = 0.07$. Additional sensitivity analyses varied $\sigma_\eta$ within $\{0.05, 0.10, 0.15\}$ to assess model robustness.

**Evaluation methodology.** Model performance was comprehensively evaluated using three criteria: (1) classification score (CS) assessing sequence-level predictive accuracy, (2) uncertainty calibration, quantified through the median and 95% confidence interval of the estimated event rate $\hat{\lambda}$, obtained via nonparametric resampling within each frequency band, and (3) structural fidelity, measured using intersection-over-union (IoU) between the predicted latent structure and the ground-truth event patterns.

### F.2.2 Alzheimer's Disease EEG Dataset

**Dataset AD Cohort A** (Miltiadous et al., 2024) consists of resting-state, eyes-closed EEG recordings from a total of 88 participants, categorized into 36 individuals diagnosed with Alzheimer's disease (AD), 23 patients with frontotemporal dementia (FTD), and 29 healthy control subjects (HC). The EEG data were collected using 19 electrodes arranged according to the international 10–20 placement system. The recordings have a sampling rate of 500 Hz and an average duration ranging from approximately 12 to 14 minutes per subject. Provided in adherence to the Brain Imaging Data Structure (BIDS) standard, the dataset includes both raw and preprocessed EEG signals, enabling robust comparative analysis across different dementia subtypes.

**Dataset AD Cohort B** (Sadegh-Zadeh et al., 2023) includes resting-state EEG data from 168 participants, segmented into 59 moderate Alzheimer's disease patients (AD), 7 individuals diagnosed with mild cognitive impairment (MCI), and 102 healthy controls (HC). EEG recordings were acquired using the standardized 10–20 electrode placement system, with data presented in MATLAB (.mat) format. Accompanying the EEG data are Mini-Mental State Examination (MMSE) scores, providing cognitive assessments for participants. This dataset is particularly tailored for the distinction of AD from MCI, thus serving as a valuable resource for investigations aimed at early Alzheimer's disease diagnosis.

### F.3 Experimental Setup

We evaluate BayesENDS under a five–fold cross–subject protocol to ensure that generalization is assessed on previously unseen participants rather than unseen windows from the same individuals. For each cohort, subjects are partitioned into five disjoint folds with stratification at the subject level so that the class proportions of Alzheimer's disease, frontotemporal dementia/mild cognitive impairment, and healthy controls are approximately preserved in every split. In each round, four folds are used for training, and one for testing; the roles of the folds are rotated until every fold serves exactly once in the held–out test set.

EEG is segmented into non–overlapping two–second windows (1,000 samples at 500 Hz) per subject and channel, followed by channel–wise $z$–normalization computed within the training portion of the active fold and then applied to validation and test windows of that fold.

Evaluation is conducted at subject levels. Subject–level predictions aggregate a subject's windows by majority vote over window–wise labels. We report accuracy, macro–F1, and summarize performance as the mean and standard deviation across the five test folds.

## G Architectures and Training Details

This section gives a concise description of the components used in BayesENDS and the training protocol.

### G.1 INPUT, PREPROCESSING, AND WINDOWING

EEG is segmented into non–overlapping windows and $z$–scored channel–wise using statistics computed on the training split of each fold. In our AD experiments we use 2,s windows from 500,Hz recordings ($T{=}1000$) and $C{=}19$ electrodes (10–20 layout). Other datasets can adjust $C$ and $T$ without changing the architecture.

### G.2 ENCODER

The encoder follows the temporal–spatial factorization popular in EEGNet, with a mild max–norm constraint on spatial depthwise kernels for stability on EEG.

- **Block 1 (temporal → spatial).** Depthwise temporal convolution → BatchNorm → ELU; then depthwise *spatial* convolution across electrodes → BatchNorm → ELU; time average pooling; dropout (0.1).

- **Block 2 (depthwise–separable temporal).** Depthwise temporal convolution → BatchNorm → ELU; pointwise mixing → BatchNorm → ELU; time average pooling; dropout (0.1).

- **Two branches.** (a) A flattened *main* feature vector is used by the classifier; (b) a temporally downsampled, per–electrode feature map feeds the EPDE/MELP/dLIF block.

### G.3 EPDE + MELP + DLIF COUPLING (LATENT EVENT DYNAMICS)

Given the encoder's per–electrode temporal features, the EPDE produces a differentiable posterior over next–event times. We parameterize a small MLP per channel to output mixture parameters for the **MELP** (lognormal mixture; $K{=}3$ components). Sampling is reparameterized during training; at test time we use mixture expectations. A compact hidden state is evolved with an explicit–Euler solver to obtain a denoised per–electrode trajectory used downstream. A rate proxy read from the EPDE is softly aligned with the **dLIF** prior via an $L_2$ rate consistency term with refractory gating and a plausible alpha/theta–to–beta frequency range.

### G.4 EVENT–RELATIONAL GRAPH (ERG) AND GCN

From posterior samples of event times, cross–channel lags are mapped through a smooth STDP–shaped nonlinearity to edge scores in $[0, 1]$, then averaged over time/samples to produce a symmetric adjacency. A weak Fisher–$z$ alignment term biases edges toward observable correlations computed from the raw EEG without enforcing them. A single GCN layer converts per–channel temporal descriptors into compact node embeddings that are flattened for fusion.

### G.5 CLASSIFIER AND FUSION

We concatenate the encoder's main vector with the flattened GCN features and apply a two–layer MLP followed by a linear layer and Softmax over classes. No attention or recurrence is used at this stage; temporal information is already summarized by EPDE/MELP and the encoder.

### G.6 OPTIMIZATION AND PROTOCOL

- **Loss.** Cross–entropy for labels plus small auxiliary terms: EPDE/MELP reconstruction/regularizers, dLIF rate consistency, ERG Fisher–$z$, and the IVP–KL surrogate. We use modest default weights and found them robust across cohorts.

- **Training.** Adam (lr $5{\times}10^{-4}$, weight decay $10^{-4}$), batch size 1024, gradient–norm clipping at 1.0, 30 epochs. Learning rate is halved if validation AUC does not improve for 15 epochs; the best AUC checkpoint is kept.

- **Evaluation.** Five–fold *cross–subject* splits; subject–level predictions from window probabilities via majority vote (or simple averaging).

## G.7 SHAPES SUMMARY

Below we list only input/output sizes and activations for clarity; $L{=}250$ denotes the temporal length after the first time–pooling stage.

| Module | Input | Output | Activation |
|---|---|---|---|
| Input window | $\mathbb{R}^{B \times 1 \times 19 \times 1000}$ | – | – |
| Encoder (temporal branch) | $\mathbb{R}^{B \times 1 \times 19 \times 1000}$ | $\mathbb{R}^{B \times 19 \times L}$ $(L=250)$ | ELU |
| Encoder (main branch, flattened) | $\mathbb{R}^{B \times 1 \times 19 \times 1000}$ | $\mathbb{R}^{B \times 1178}$ | ELU |
| EPDE/MELP/dLIF block | $\mathbb{R}^{B \times 19 \times L}$ | $\mathbb{R}^{B \times 19 \times L}$ | ELU (MLPs), Tanh (ODE) |
| ERG adjacency | lags from EPDE/MELP | $\mathbb{R}^{B \times 19 \times 19}$ | exp kernel |
| GCN node embeddings | $\mathbb{R}^{B \times 19 \times L}$, adjacency | $\mathbb{R}^{B \times 19 \times 64}$ | ReLU |
| Flattened graph features | $\mathbb{R}^{B \times 19 \times 64}$ | $\mathbb{R}^{B \times 1216}$ | – |
| Fusion vector | concat(main, graph) | $\mathbb{R}^{B \times 2394}$ | – |
| Classifier logits | $\mathbb{R}^{B \times 2394}$ | $\mathbb{R}^{B \times |\mathcal{Y}|}$ | ReLU (hidden), Softmax (out) |

## G.8 PSEUDOCODE SUMMARIES

---

**Algorithm 1** MELP Sampling (per electrode and time proxy)

---

**Require:** Mixture weights $w \in \Delta^{K-1}$, means $\mu \in \mathbb{R}^K$, standard deviations $\sigma \in \mathbb{R}_+^K$
**Ensure:** Inter-event interval $\tau \in \mathbb{R}_+$
1: Sample component $k \sim \text{Categorical}(w)$
2: Sample noise $\epsilon \sim \mathcal{N}(0,1)$
3: $\tau \leftarrow \exp\big(\mu_k + \sigma_k \cdot \epsilon\big)$
4: **return** $\tau$

---

---

**Algorithm 2** Neural ODE evolution over $[t_s, t_e]$ with $S$ Euler sub-steps

---

**Require:** Features $\xi$, projection map $\text{proj}(\cdot)$, decoder $\text{decode}(\cdot)$, vector field $f(\cdot)$, residual weight $\alpha > 0$, sub-steps $S \in \mathbb{N}$, interval $[t_s, t_e]$
**Ensure:** Decoded trajectory $\hat{z} \in \mathbb{R}^{\lfloor T/4 \rfloor}$
1: $y_0 \leftarrow \text{proj}(\xi)$
2: $\Delta t \leftarrow (t_e - t_s)/S$
3: **for** $m = 0, 1, \ldots, S-1$ **do**
4: $\quad y_{m+1} \leftarrow y_m + \Delta t \cdot \big(f(y_m) + \alpha\, y_m\big)$
5: **end for**
6: $\hat{z} \leftarrow \text{decode}(y_S)$
7: **return** $\hat{z}$

---

---

**Algorithm 3** Graph weights from $z_t$

---

**Require:** Per-electrode trajectories $z_t \in \mathbb{R}^{N \times L}$ with $L = \lfloor T/4 \rfloor$; scale $\gamma > 0$; kernel mode mode $\in \{\texttt{exp}, \texttt{gauss}, \texttt{inv1}\}$
**Ensure:** Symmetric adjacency $W \in \mathbb{R}^{N \times N}$ with zero diagonal

1: $\forall c \in \{1, \ldots, N\}: s_c \leftarrow \frac{1}{L} \sum_{t=1}^{L} z_t(c, t)$
2: **for** $i = 1, \ldots, N$ **do**
3:     **for** $j = 1, \ldots, N$ **do**
4:         **if** $i = j$ **then**
5:             $W_{ij} \leftarrow 0$
6:         **else**
7:             $\Delta \leftarrow |s_i - s_j|$
8:             $W_{ij} \leftarrow \exp(-\gamma \, \Delta)$
9:         **end if**
10:     **end for**
11: **end for**
12: $W \leftarrow \frac{1}{2}(W + W^\top)$                          $\triangleright$ Enforce symmetry
13: **return** $W$

---

**Algorithm 4** BayesENDS: Training and Inference

---

**Require:** Dataset $\mathcal{D} = \{(X, Y)\}$, electrodes $C$, window length $T$, MELP comps $K$, ODE substeps $S$
**Require:** Loss weights $\lambda_{\text{aux}}, \lambda_{\text{spk}}, \lambda_{\text{graph}}, \lambda_{\text{LIF}}$; LR $\eta$
**Ensure:** Trained params $\Theta$; predictor BayesENDS$(\cdot)$

1: Initialize encoder $\psi$, EPDE+MELP $\phi$, dLIF head $\xi$, ERG+GCN $\eta_g$, classifier $\theta$; Adam$(\eta)$
2: **for** epoch $= 1..E$ **do**
3:     **for** mini-batch $(X, Y) \sim \mathcal{D}$ **do**
4:         **Preprocess**: channel-wise $z$-score per window
5:         **Encoder** ($\psi$): $\texttt{main\_vec} \in \mathbb{R}^{B \times d_{\text{main}}}$, $\texttt{temp\_feat} \in \mathbb{R}^{B \times C \times 1 \times L}$, $L = \lfloor T/4 \rfloor$
6:         **EPDE+MELP+ODE**:
      For $c = 1..C$: EPDE $\Rightarrow$ mixture $(\mathbf{w}_c, \boldsymbol{\mu}_c, \boldsymbol{\sigma}_c)$; sample $\tau = \exp(\mu_{c,k} + \sigma_{c,k}\varepsilon)$, accumulate events $\{T_c\}$; ODE evolve with $S$ Euler steps $\Rightarrow$ trajectory $z_{c,:}$
7:         Stack $Z \in \mathbb{R}^{B \times C \times L}$; keep event lags for ERG
8:         **dLIF prior**: compute $r_c(t)$ from $Z$ with refractory gate; rate proxy $\widehat{r}_c(t)$
9:         **Regularizers**: $\mathcal{R}_{\text{LIF}} = \sum_c \int (\widehat{r}_c - r_c)^2$;    $\text{KL}_T = \sum_c \mathcal{U}_\varepsilon(q_c \| p_{\text{dLIF}})$ (IVP–KL bound)
10:         **ERG** from event lags: $e_{ij}(t) = \exp(-\alpha |\Delta \tilde{t}_{ij}(t)|)$, average $\Rightarrow \bar{A}$
11:         **Fisher–$z$ alignment**: $\mathcal{R}_{\text{ERG}} = \sum_{i<j} \left[ (z_{ij}^{\text{obs}} - z_{ij}^{\text{pred}})^2 / (2\sigma_{ij}^2) + \frac{1}{2} \log \sigma_{ij}^2 \right]$
12:         **GCN** on $(\bar{A}, Z)$: $G \in \mathbb{R}^{B \times (C \cdot d_g)}$;    **Fuse** $H = [\texttt{main\_vec}; G]$
13:         **Classifier**: logits $\ell = \text{MLP}(H)$, $p = \text{softmax}(\ell)$
14:         **Loss**: $\mathcal{L} = \text{CE}(p, Y) + \lambda_{\text{aux}}\mathcal{L}_{\text{STRODE}} + \lambda_{\text{spk}}\mathcal{L}_{\text{spk}} + \lambda_{\text{graph}}\mathcal{R}_{\text{ERG}} + \lambda_{\text{LIF}}\mathcal{R}_{\text{LIF}} + \text{KL}_T$
15:         Backprop; clip $\|\nabla\| \leq 1$; Adam step on $\Theta$
16:     **end for**
17: **end for**

18: **Inference on window** $X$:
19: Preprocess $\rightarrow$ Encoder; EPDE gives MELP expectations $\rightarrow$ events $\{T_c\}$; ODE $\rightarrow Z$; build $\bar{A}$; GCN $\rightarrow G$; fuse $\rightarrow H$; output $p$ and $\hat{y} = \arg\max p$
20: **Subject aggregation**: majority vote $\Rightarrow \hat{y}_{\text{subj}}$

---

