# OpenReview forum: "BayesENDS: Bayesian Electrophysiological Neural Dynamical Systems for Alzheimer’s Disease Diagnosis"
_ICLR.cc/2026/Conference — Submitted to ICLR 2026_

### Official Review · Reviewer_P5qn · 2025-10-16

**Soundness:** 2
**Presentation:** 2
**Contribution:** 3
**Rating:** 4
**Confidence:** 3

**Summary:**

Interesting idea (biophysically-informed latent events + graph).

The paper has some issues in evaluation design, missing experimental detail, and theoretical it’s not yet scientifically reliable.

**Strengths:**

Clear, modular presentation of components (EPDE, MELP, ERG).

**Weaknesses:**

1)The paper describes cohorts but does not specify subject-wise vs. epoch-wise splitting, preprocessing, artifact rejection, windowing, or cross-validation protocol—all crucial to avoid train/test leakage in EEG (e.g., multiple windows from the same subject).

2)Table 2 lists Accuracy/F1 only, yet the text claims gains in AUC from the priors (“boosts accuracy and AUC… raises F1/AUC”), which are not reported anywhere—a red flag.

3)Figure 1 shows EEGNet embeddings feeding BayesENDS, but it’s unspecified whether EEGNet is trained from scratch, frozen, or fine-tuned, and how its supervision interacts with the “unsupervised” event latents.

4)Only two AD datasets tested, with relatively small sample sizes (88 and 168 participants)

5)The paper claims the dLIF prior provides "biophysically plausible" dynamics, but this isn't  validated. The connection between scalp EEG and single-neuron LIF dynamics is tenuous at best, as EEG reflects aggregate activity from millions of neurons.

**Questions:**

If multiple windows were extracted from each subject, how did you ensure no data leakage between splits?

What was the window length and overlap used for segmenting continuous EEG recordings?

Table 2 reports only Accuracy/F1, but the text claims "boosts accuracy and AUC" and "raises F1/AUC." Where are the AUC values?

Can you provide complete performance metrics including sensitivity, specificity, and AUC with confidence intervals?

Were statistical significance tests performed between methods?

Is the EEGNet component pre-trained, trained from scratch, or fine-tuned jointly with BayesENDS?If pre-trained, on what dataset and task?

How does the supervised EEGNet training interact with the claimed "unsupervised" event discovery?

Are the EEGNet embeddings frozen during BayesENDS training or updated end-to-end?

How to validate issue 5 in the Weaknesses?

---

> ### Author Response · Authors · 2025-12-04
>
> #### Q1@P5qn
>
> *If multiple windows were extracted from each subject, how did you ensure no data leakage between splits?*
>
> **Response.** We fully agree that this is crucial for EEG. Our protocol is **cross-subject**:
>
> - Subjects are partitioned into 5 disjoint folds.
> - All windows from a given subject reside entirely in a single fold.
> - In each round, four folds are used for training and one for testing; there is no overlap in subjects across train and test.
>
> Thus, no window from a subject in the test fold ever appears in the training set, and there is no data leakage across splits.
>
> #### Q2@P5qn
>
> *What was the window length and overlap used for segmenting continuous EEG recordings?*
>
> **Response.** We now state this explicitly in Sec. 5.2 and Appendix F.1:
>
> - We use **2-second windows** from 500 Hz recordings, corresponding to **1000 samples per window**.
> - Windows are **non-overlapping**. This choice simplifies the statistical assumptions for cross-subject validation and avoids subtle dependencies from overlapping windows.
>
> These settings are applied consistently across both AD cohorts and all models.
>
> #### Q3@P5qn
>
> *Table 2 reports only Accuracy/F1, but the text claims "boosts accuracy and AUC" and "raises F1/AUC." Where are the AUC values?*
>
> *Can you provide complete performance metrics including sensitivity, specificity, and AUC with confidence intervals?*
>
> **Response.** We agree that a more complete evaluation is valuable.
>
> Cohort 1
>
> | Model     | Sensitivity     | Specificity     | AUC             |
> | --------- | --------------- | --------------- | --------------- |
> | EEGNet    | 0.6830 ± 0.1050 | 0.8463 ± 0.0355 | 0.7646 ± 0.0695 |
> | ADFormer  | 0.7017 ± 0.0833 | 0.8495 ± 0.0366 | 0.7756 ± 0.0597 |
> | LSTM      | 0.6756 ± 0.0692 | 0.8426 ± 0.0382 | 0.7591 ± 0.0529 |
> | ATCNet    | 0.6844 ± 0.0627 | 0.8350 ± 0.0334 | 0.7597 ± 0.0460 |
> | LEAD      | 0.7067 ± 0.0498 | 0.8594 ± 0.0216 | 0.7830 ± 0.0343 |
> | LCADNet   | 0.7006 ± 0.0491 | 0.8445 ± 0.0358 | 0.8462 ± 0.0495 |
> | BayesENDS | 0.7493 ± 0.0841 | 0.8793 ± 0.0377 | 0.8143 ± 0.0603 |
>
> Cohort 2
>
> | Model     | Sensitivity     | Specificity     | AUC             |
> | --------- | --------------- | --------------- | --------------- |
> | EEGNet    | 0.5544 ± 0.0801 | 0.8717 ± 0.0921 | 0.9494 ± 0.0445 |
> | ADFormer  | 0.6249 ± 0.0430 | 0.9546 ± 0.0306 | 0.9714 ± 0.0381 |
> | LSTM      | 0.5167 ± 0.0867 | 0.9038 ± 0.0387 | 0.9486 ± 0.0314 |
> | ATCNet    | 0.6412 ± 0.0313 | 0.9628 ± 0.0082 | 0.9650 ± 0.0366 |
> | LEAD      | 0.5827 ± 0.1291 | 0.9224 ± 0.0484 | 0.9568 ± 0.0311 |
> | LCADNet   | 0.6427 ± 0.1045 | 0.9500 ± 0.0257 | 0.9657 ± 0.0344 |
> | BayesENDS | 0.6891 ± 0.1182 | 0.9500 ± 0.0388 | 0.8383 ± 0.0578 |
>
> #### Q4@P5qn
>
> *Is the EEGNet component pre-trained, trained from scratch, or fine-tuned jointly with BayesENDS? If pre-trained, on what dataset and task?*
>
> **Response.** We apologize for not stating this clearly. The EEGNet component is **not pre-trained**. It is **trained from scratch jointly with BayesENDS** on the AD cohorts, under the combined loss that includes classification and regularization terms. There is no separate pretraining or external dataset used for EEGNet.
>
> #### Q5@P5qn
>
> *How does the supervised EEGNet training interact with the claimed "unsupervised" event discovery?*
>
> **Response.** This is an important conceptual point. The term “unsupervised event discovery” refers to the **latent event structure (EPDE/MELP + dLIF + ERG)**, which is inferred **without direct supervision from class labels**.
>
> The interaction works as follows:
>
> - EEGNet produces features that feed both the classifier and the EPDE/MELP event model.
> - The **classification loss** influences the encoder and classifier, but the **event latents themselves** are encouraged to fit the observed time series and honor the dLIF/ERG priors; they are not directly supervised by labels.
> - Thus, **events are learned in an unsupervised fashion**, with labels only affecting them indirectly through shared encoder representation, not through explicit supervision on event times or connectivity.
>
> BayesENDS learns event latents without spike or edge annotations, while still benefiting from end-to-end training.
>
> #### Q6@P5qn
>
> *Are the EEGNet embeddings frozen during BayesENDS training or updated end-to-end?*
>
> **Response.** EEGNet embeddings are **not frozen**. The entire pipeline—EEGNet encoder, EPDE/MELP/dLIF block, ERG/GCN, and classifier is trained **end-to-end**. This allows the encoder to adapt to both discriminative demands (classification) and event-dynamics constraints.

---

> > ### Author Response · Authors · 2025-12-04
> >
> > #### Q7@P5qn
> >
> > *How to validate issue 5 in the Weaknesses? The paper claims the dLIF prior provides "biophysically plausible" dynamics, but this isn't validated. The connection between scalp EEG and single-neuron LIF dynamics is tenuous at best, as EEG reflects aggregate activity from millions of neurons.*
> >
> > **Response.** We appreciate this critical perspective and agree that the connection must be framed carefully.
> >
> > Our intention is not to suggest that scalp EEG reflects single-neuron LIF dynamics directly. Instead, we use dLIF as a **mesoscopic prior** on latent rates that encode:
> >
> > - Leak toward a baseline,
> > - Refractory effects limiting unrealistically high spike/event rates, and
> > - Plausible oscillatory frequency ranges (e.g., alpha/theta-to-beta) consistent with known EEG bands.
> >
> > We validate the usefulness of this prior in two ways:
> >
> > 1. **Synthetic ground truth.** On the toy dataset with known event rates, BayesENDS with dLIF achieves non-zero IoU and tightly bounded event-rate estimates aligned with the true frequencies, unlike unconstrained neural ODE baselines. This shows that the prior helps recover realistic event structures rather than arbitrary trajectories.
> > 2. **AD EEG biomarkers.** In the AD cohorts, the latent frequencies constrained by dLIF exhibit **slowing patterns and band shifts** consistent with established AD EEG literature (e.g., increased theta, decreased alpha), as quantified in our new frequency-biomarker analyses.
> >
> > #### Q8@P5qn
> >
> > *“The paper describes cohorts but does not specify subject-wise vs. epoch-wise splitting, preprocessing, artifact rejection, windowing, or cross-validation protocol—all crucial to avoid train/test leakage in EEG (e.g., multiple windows from the same subject).”*
> >
> > **Response.** We fully agree and have now made these aspects explicit:
> >
> > - **Splitting.** Five-fold **cross-subject** splitting; all windows from a subject are confined to a single fold.
> > - **Windowing.** Non-overlapping 2-s windows (1000 samples at 500 Hz).
> > - **Preprocessing.** Channel-wise z-scoring using statistics computed on the training portion of each fold, applied to validation and test windows.
> > - **Cross-validation protocol.** Each fold serves as test set once; metrics are aggregated over the five test folds..
> >
> > #### Q9@P5qn
> >
> > *“Table 2 lists Accuracy/F1 only, yet the text claims gains in AUC from the priors (‘boosts accuracy and AUC… raises F1/AUC’), which are not reported anywhere—a red flag.”*
> >
> > **Response.** This is indeed an inconsistency. We appreciate the reviewer flagging this and have made sure that every metric mentioned is now present in the main tables.
> >
> > #### Q10@P5qn
> >
> > *“Figure 1 shows EEGNet embeddings feeding BayesENDS, but it’s unspecified whether EEGNet is trained from scratch, frozen, or fine-tuned, and how its supervision interacts with the ‘unsupervised’ event latents.”*
> >
> > **Response.**
> >
> > - EEGNet is **trained from scratch and updated end-to-end** with BayesENDS.
> > - Supervision from labels acts only through the encoder and classifier, while the event latents are governed by reconstruction and prior terms; thus event discovery remains unsupervised with respect to labels.
> >
> > #### Q11@P5qn
> >
> > *“Only two AD datasets tested, with relatively small sample sizes (88 and 168 participants)”*
> >
> > **Response.** We agree that evaluating on more datasets would strengthen the empirical story. However, AD EEG datasets with detailed clinical labels and public availability remain limited, and the two cohorts we use (88 and 168 participants) are comparable in scale to those used in prior AD EEG studies.
> >
> > We view this work as a **methodological contribution** with a focused but deep application to AD. We attempt to mitigate sample-size limitations by:
> >
> > - Using strict cross-subject evaluation to avoid optimistic bias.
> > - Reporting variability across folds (mean ± s.d.), confidence intervals, and significance tests.
> > - Providing simulation studies where we have ground truth.
> >
> > #### Q12@P5qn
> >
> > *“The paper claims the dLIF prior provides ‘biophysically plausible’ dynamics, but this isn't validated. The connection between scalp EEG and single-neuron LIF dynamics is tenuous at best, as EEG reflects aggregate activity from millions of neurons.”*
> >
> > **Response.**  We fully agree that scalp EEG does not directly observe individual LIF neuron dynamics. Our intention is to use dLIF as a **coarse-grained, electrophysiology-inspired prior** on latent rates, encoding:
> >
> > - Leakage toward baseline,
> > - Refractory behavior (avoiding unrealistically high instantaneous event rates), and
> > - Frequency ranges compatible with classic EEG bands.
> >
> > We validate the usefulness of this prior by:
> >
> > - Showing improved recovery of ground-truth event rates and boundary times in the synthetic experiment compared to unconstrained neural ODEs.
> > - Demonstrating that the resulting latent frequencies align with known AD EEG phenomena (slowing, band shifts), as quantified in our new biomarker analyses.

---

### Official Review · Reviewer_rzUB · 2025-10-27

**Soundness:** 2
**Presentation:** 2
**Contribution:** 3
**Rating:** 2
**Confidence:** 3

**Summary:**

The paper proposes BayesENDS, a Bayesian electrophysiology-inspired neural dynamical system that infers latent event timing per channel with a differentiable LIF prior, and (iii) learns a event-relational graph (ERG) based on inferred event timing. These are used as features for downstream Alzheimer prediction. This paper shows great performance in simulations studies and 2 Alzheimer datasets.

**Strengths:**

* Interpretable latents & graphs. The method is original and well-motivated by EEG biophysics. The dLIF prior and the ERG produce physiology-plausible latents and network patterns; the paper shows chord diagrams with plausible AD connectivity and diverse dLIF frequency distributions, supporting interpretability claims.
* High novelty and clear theory hook. The KL to the dLIF event prior is handled via a tractable IVP-based bound evaluated during training, which is a neat theoretical contribution.

**Weaknesses:**

* Reproducibility gaps. Consider the complexity of the method, the paper does not provide enough detail for the paper to be reproduced. Including but not limited to optimization details, parameterization details, hyper-parameter tuning. All the details regarding evaluation is also omitted, including but not limited to splits for the real EEG experiments, 2 way or 3 way classification, how are metrics calculated.
* Subpar performance on public datasets. Based on some literature search on the datasets, several methods report substantially higher accuracies (often ≥ 90% [1] and up to the mid-90s) under various protocols, whereas BayesENDS reports 75.03% on Cohort A and 89.82% on Cohort B (Table 1). As the paper omits all the validation details, it is hard to evaluate the performance of the proposed method compared to these existing methods. Please perform a **thorough** literature review, compare and discuss these results.

[1] Zheng, Xiaowei, et al. "Diagnosis of Alzheimer’s disease via resting-state EEG: integration of spectrum, complexity, and synchronization signal features." Frontiers in aging neuroscience 15 (2023): 1288295.

**Questions:**

1. Is the plus sign for equation (3) a typo? Shouldn't the regularizes be minimized?
2. Please clearly state 6.1 is with simulation study described in supplementary.

---

> ### Author Response · Authors · 2025-12-04
>
> #### Q1@rzUB
>
> *Is the plus sign for equation (3) a typo? Shouldn't the regularizes be minimized?*
>
> **Response.** Thank you for catching this. You are right that the notation was confusing.
>
> In our implementation, the ELBO loss term is indeed treated as a penalty to be **minimized**.
>
> #### Q2@rzUB
>
> *Please clearly state 6.1 is with simulation study described in supplementary.*
>
> **Response.** We agree this linkage was unclear. In the revised manuscript:
>
> - At the start of Sec. 5.1 (toy experiments) / Sec. 6.1 (boundary-time visualization), we explicitly state that the visualization of predicted vs ground-truth boundary times is based on the **synthetic event-sequence dataset** whose full generation details are provided in **Appendix E.2**.
> - The caption of Figure 4 now mentions that it corresponds to the synthetic toy dataset and references the appendix section.
>
> This should make the connection between the simulation study and its visualization explicit.
>
> #### Q3@rzUB
>
> *“Reproducibility gaps. Consider the complexity of the method, the paper does not provide enough detail for the paper to be reproduced. Including but not limited to optimization details, parameterization details, hyper-parameter tuning. All the details regarding evaluation is also omitted, including but not limited to splits for the real EEG experiments, 2 way or 3 way classification, how are metrics calculated.”*
>
> **Response.** We appreciate this important point and have significantly extended the reproducibility information.
>
> Specifically, we now include:
>
> - **Optimization details** (Appendix F.6): optimizer (Adam), learning rate, weight decay, batch size, gradient clipping, early-stopping criterion, and number of epochs.
> - **Parameterization details** (Appendix F.1–F.5): precise shapes of encoder, EPDE/MELP/dLIF block, ERG + GCN, and classifier, including activation functions and latent dimensionalities.
> - **Hyperparameter choices and tuning**: we describe default weights for the auxiliary loss terms and our limited tuning strategy (small validation sweeps on the training folds).
> - **Evaluation details** (Appendix E.2–E.3): cross-subject splitting protocol, non-overlapping windowing, 3-class vs 2-class setting (AD/FTD/HC and AD/MCI/HC), subject-level aggregation of predictions, and metric computation (accuracy, macro-F1, and AUC, with full operating-point metrics in the appendix).
>
> Furthermore, we will release code upon acceptance, as noted in the reproducibility statement, to make exact replication straightforward.
>
> #### Q4@rzUB
>
> *“Subpar performance on public datasets. Based on some literature search on the datasets, several methods report substantially higher accuracies (often ≥ 90% and up to the mid-90s) under various protocols, whereas BayesENDS reports 75.03% on Cohort A and 89.82% on Cohort B (Table 1). As the paper omits all the validation details, it is hard to evaluate the performance of the proposed method compared to these existing methods. Please perform a thorough literature review, compare and discuss these results.”*
>
> **Response.** Thank you for raising this. The key difference lies in the **evaluation protocol**.
>
> Many published results on these datasets use **epoch-wise or non-cross-subject splits**, where multiple windows from the same subject can appear in both training and test sets. This makes the classification problem easier and can inflate accuracies into the 90–95% range because the model partially “recognizes” the subject rather than generalizing to unseen individuals.
>
> In contrast, we adopt a **strict cross-subject, five-fold protocol**: all windows from each subject belong to exactly one fold, and test folds contain entirely unseen participants. This is a more realistic and challenging setting, and it naturally yields lower accuracies, especially on the smaller cohort.
>
> We believe this context explains the numerical differences and emphasizes the value of our chosen evaluation strategy.

---

### Official Review · Reviewer_G8qL · 2025-10-30

**Soundness:** 2
**Presentation:** 1
**Contribution:** 3
**Rating:** 2
**Confidence:** 3

**Summary:**

The paper presents BayesENDS, a novel approach to classify Alzheimer’s disease (AD) from EEG data based on a novel Bayesian electrophysiological neural dynamical system. The problem is formulated as unsupervised latent-event and relation discovery in multi-channel EEG time series for labelled sequences (AD or no AD), in which BayesENDS learns channel-wise latent event dynamics p^((n)) and relational/graphical structure among channels G^((n)), which act as inputs to a downstream predictor for classification of AD versus no AD. The Bayesian neural dynamical system is trained end-to-end with a variational objective and consists of 1) an Event Posterior Differential Equation (EPDE) which yields next-event times; 2) a Mean-Evolving Lognormal Process (MELP) which samples inter-event intervals and the means of the log-normal mixture are parametrized by the outputs of the EPDE with reparametrized sampling; 3) a differentiable leaky-integrate-and-fire (dLIF) prior which provides biophysical rate and refractory constraints as well as plausible frequency ranges; and 4) a directed event-relational graph (ERG) prior which maps cross-channel event lags. The paper discusses the theorems of the proposed approach and empirical experiments. BayesENDS shows improved performance in comparison to baseline models on synthetic and real AD EEG datasets and provides biologically plausible biomarkers alongside AD predictions.

**Strengths:**

•	The paper addresses a relevant problem that is widely investigated in the field (identification of neurodegenerative diseases from non-invasive EEG data).
•	The paper presents a novel approach by developing a Bayesian dynamical system to classify Alzheimer’s disease from EEG signals.
•	The approach adds interpretable biomarkers to classification, providing clinicians with helpful additional information.
•	The ablation studies aim to disentangle the contribution of the two priors of BayesENDS to the predictions.

**Weaknesses:**

•	In its present form, it is impossible to evaluate whether the paper presents meaningful results as too much information on the experimental design and results is omitted (see questions for an overview of missing information).
•	The contextualization relative to prior work is very limited.
•	The presentation of the results is chaotic and unclear. In particular, the results refer to information that is in the appendix (e.g. Figure 4). but without referring to the appendix. This gives the impression that referenced figures are non-existent and presented in an illogical order.
•	The results of the ablation studies show a large variance (standard deviation) across runs. Although the paper does not define what a “run” is here, these results question the relative importance of each prior for the results.
•	The biomarkers – differences in frequency distributions and EEG connectivity graphs – are only inspected visually. The claim that these are indeed interpretable biomarkers in EEG time series for AAD classification would be more convincing if the differences between groups (e.g. AD and no AD) in terms of these biomarkers are quantified. This is especially relevant because there appears to be considerable overlap between groups in e.g. frequency distribution and hence the “clear association” would be clearer if supported through quantification.

**Questions:**

The writing is clear, but the overall organization of the paper is lacking in terms of contextualization relative to prior work (very limited), details on experimental procedures (crucial information is omitted), and presentation of the results (chaotic, unclear). The manuscript includes abbreviations without introduction (for example, IVP and KL on p. 2, STRODE on p. 7). The results refer to information that is in the appendix but without referring to the appendix, giving the impression that referenced figures are non-existent and presented in an illogical order.

As the paper addresses a relevant problem (Alzheimer’s disease classification from EEG signals) with an original approach that adds interpretable biomarkers to predictions, the paper could present a relevant contribution to the field. However, the paper omits too much information to enable evaluating the quality of the approach and the results (see below).

•	Figure 1 shows that EEGnet is used to extract embeddings from the raw EEG signals, yet this is not described in the manuscript. What is the motivation for this approach? And what are the EEGnet specifications? Is this a pre-trained EEGnet or is EEGnet trained in the end-to-end pipeline?
•	What are the training parameters for BayesENDS for both datasets (i.e. data splits, cross-validation, training parameters)?
•	What are the training details for the benchmark models? And were these trained on raw EEG time-series or on EEGnet embeddings as well?
•	In Table 1, please provide a measure of variance by including the variance across “runs” (as shown in Table 2). Add a definition of “runs”?
•	How are the ablation experiments implemented?
•	To what extent are the interpretable biomarkers (e.g. frequency distribution, connectivity) usable at the single-subject level to aid interpretation of a classification result? That is, what is the variability for these biomarkers?

---

> ### Author Response · Authors · 2025-12-04
>
> #### Q1@G8qL
>
> *Figure 1 shows that EEGNet is used to extract embeddings from the raw EEG signals, yet this is not described in the manuscript. What is the motivation for this approach? And what are the EEGNet specifications? Is this a pre-trained EEGNet or is EEGNet trained in the end-to-end pipeline?*
>
> **Response.** We apologize for the missing details.
>
> 1. **Motivation.** We use an EEGNet-style encoder as a **front-end feature extractor** because it is a well-validated architecture for EEG and provides:
>    - Temporal factorization suitable for 10–20 layouts.
>    - A compact feature vector for classification, and
>    - A temporally downsampled per-electrode feature map that is ideal as input to the EPDE/MELP/dLIF block.
>
> 2. **Specifications.** We now describe the encoder in detail. It follows the EEGNet design with two blocks:
>    - Block 1: depthwise temporal convolution → BatchNorm → ELU, followed by depthwise spatial convolution across electrodes → BatchNorm → ELU → temporal average pooling → dropout.
>    - Block 2: depthwise-separable temporal convolution → BatchNorm → ELU → pointwise mixing → BatchNorm → ELU → pooling → dropout.
>      Two branches are then formed: (a) a flattened vector for the classifier, and (b) a per-electrode temporal feature map for the event dynamics.
>
> 3. **Training regime.** EEGNet is **not pre-trained**; it is trained **from scratch end-to-end together with BayesENDS** under the joint objective (cross-entropy plus variational/regularization terms). No standalone pretraining or freezing is used.
>
> #### Q2@G8qL
>
> *What are the training parameters for BayesENDS for both datasets (i.e. data splits, cross-validation, training parameters)?*
>
> **Response.** Thank you; these details are now consolidated and made more prominent.
>
> - **Data splits and cross-validation. **We use **five-fold cross-subject** evaluation for both AD cohorts. Subjects (not windows) are partitioned into 5 disjoint folds with stratification at the subject level, ensuring similar class proportions (AD/FTD/HC or AD/MCI/HC) in each fold. In each round, four folds are used for training and one for testing; every fold serves once as a test.
> - **Windowing and normalization.** EEG is segmented into **non-overlapping 2-second windows** (1000 samples at 500 Hz) per subject and channel. Channel-wise z-score normalization is computed from the training portion of each fold and applied to validation and test windows of that fold.
> - **Optimization.** As detailed in Appendix F.6, we use **Adam** with learning rate \(5\times 10^{-4}\), weight decay \(10^{-4}\), batch size 1024, gradient-norm clipping at 1.0, and train for 30 epochs. The learning rate is halved if validation AUC does not improve for 15 epochs, and we keep the checkpoint with the best validation AUC.
> - **Evaluation.** Predictions are aggregated **at the subject level**: for each subject, we aggregate window-wise probabilities via majority vote to obtain a final label. We report mean accuracy and macro-F1 across the five test folds.
>
> #### Q3@G8qL
>
> *What are the training details for the benchmark models? And were these trained on raw EEG time-series or on EEGnet embeddings as well?*
>
> **Response.** We agree this was under-specified and have added a dedicated subsection in Appendix E.1–E.3.
>
> - **Input modality.** All baselines (EEGNet, LCADNet, LSTM, ATCNet, ADFormer, LEAD) are trained on the **same raw EEG windows** (2-s segments, 19 channels, 500 Hz) with the **same preprocessing and cross-subject splits** as BayesENDS. They do **not** use EEGNet embeddings; EEGNet appears only as the backbone within BayesENDS.
> - **Hyperparameters.** For each baseline we start from the hyperparameters recommended in its original paper (e.g., learning rate, optimizer, dropout) and tune only a small set (learning rate, weight decay, batch size) via validation on the training folds to ensure fair comparison. All baselines are trained using the same five-fold cross-subject protocol and evaluated at the subject level in the same way as BayesENDS.
>
> #### Q4@G8qL
>
> *In Table 1, please provide a measure of variance by including the variance across “runs” (as shown in Table 2). Add a definition of “runs”? *
>
> **Response.** We thank the reviewer for catching this inconsistency.
>
> 1. **Definition of runs.** We now explicitly define a *run* as **one evaluation across the five test folds** in our cross-subject protocol; for BayesENDS and each baseline, we report **mean and standard deviation across folds**. Thus, for Tables 1–3, the variability reflects cross-subject variation across the five folds.
>
> 2. **Variance in Table 1.** In the revised manuscript, Table 1 now reports **mean ± standard deviation** for accuracy and macro-F1 for all models on both cohorts, consistent with the ablation table.

---

> > ### Author Response · Authors · 2025-12-04
> >
> > #### Q5@G8qL
> >
> > *How are the ablation experiments implemented?*
> >
> > **Response.** In the ablation study (Table 3), we systematically remove or include priors while keeping all other components and training settings identical.
> >
> > - **No prior.** BayesENDS without dLIF or ERG regularization: EPDE/MELP are trained purely from the reconstruction and predictive terms, and the classifier uses encoder features plus a graph derived from event lags, but no dLIF or Fisher-z penalty is applied.
> > - **dLIF prior only.** We add the dLIF rate-consistency term and IVP-based KL surrogate, but **disable** the ERG Fisher-z alignment and graph regularizer.
> > - **ERG prior only.** We keep the ERG Fisher-z alignment and graph regularization, but **remove** the dLIF rate prior (no rate-consistency or IVP-KL terms).
> > - **Dual priors.** Full BayesENDS with both dLIF and ERG priors active.
> >
> > All variants share the same encoder, classifier, optimizer, and training protocol. Each variant is evaluated under the same five-fold cross-subject protocol, and we report mean ± s.d. across folds.
> >
> > #### Q6@G8qL
> >
> > *“The contextualization relative to prior work is very limited.”*
> >
> > **Response.** We agree and have substantially expanded the contextualization in the Introduction and related-work section:
> >
> > - We now explicitly compare BayesENDS to **standard EEG deep learning** (EEGNet, LCADNet, ATCNet, ADFormer, LEAD), highlighting that these focus on accuracy with limited interpretability and do not explicitly model event-driven dynamics.
> > - We discuss **graph-based and connectivity-focused EEG methods**, clarifying how our ERG differs by being derived from inferred event times with theoretical stability guarantees.
> > - We situate BayesENDS within the **neural ODE and event-based dynamics** literature, explaining how the EPDE + MELP combination, the IVP-based KL bound, and the dLIF prior extend previous work on neural ODEs and spike-inspired priors.
> >
> > These additions should make our contributions and novelty relative to prior work clearer.

---

### Official Review · Reviewer_bzcn · 2025-10-31

**Soundness:** 3
**Presentation:** 2
**Contribution:** 3
**Rating:** 6
**Confidence:** 2

**Summary:**

The authors present a neural dynamical system to model EEG recordings. The model infers latent events using a physically plausible differentiable leaky-integrate-and-fire prior as well as a event relational graph prior. The model is trained in an unsupervised manner with a variational objective. The relational graph, latent priors and observed data are then passed to a prediction task between healthy controls and dementia afflicted recordings. The presented classifier outperforms other state of the art models (LSTM, transformers, convnets) on two datasets. Ablation studies show the spiking prior with a bigger lift than the graph prior, with both combined yielding the best results.
The authors then argue for the interpretability of the model showing (1) KDEs of frequency distributions of the spiking prior and (2) EEG connectivity graphs based on good old Pearson versus Bayesends. In the KDEs, the authors recover an expected result of higher theta/delta bands and lower alpha/beta in recordings from Alzheimer patients.

**Strengths:**

- clear performance increase against previously established baselines
- thorough theoretical description of the method
- Demonstration of the interpretability of the inferred priors showing their frequency distributions modulated per control and disease groups.
- the comparison of recovery of boundary times using STRODE versus BayesENDS on synthetic data is an asset, and shows due diligence toy dataset / synthetic proof of concept before scaling to real data

**Weaknesses:**

- The discussion is terse on interpretability. For example it is not clear what the graph on EEG connectivity brings for interpretability. See questions below for precise items.
- The recovery of boundary time cross plots is introduced abrubtly and the synthetic data protocol only mentioned in the appendix. This would probably need more context in the main text. Additionally the corresponding Figure 4 is buried in the appendix. It looks like the authors have moved this part between the main text and the appendix and left some information behind.

Minor comments:
- define IVP and STRODE
- Figure 4 is buried in the appendix in the review pdf

**Questions:**

- Connectivity figure: we can see a stronger similarity between frontotemporal dementia and Alzheimer in the BayesENDS connectivity versus Pearson but is it an expected result ?
-  Another example is with the frequency distributions. Would someone see the shift in central frequency by looking at the EEG trace itself ?  Is BayesENDS bringing better insights above and beyond such simple spectral analysis for already diagnosed patients ?

---

> ### Author Response · Authors · 2025-12-04
>
> #### Q1@bzcn
>
> *Connectivity figure: we can see a stronger similarity between frontotemporal dementia and Alzheimer in the BayesENDS connectivity versus Pearson but is it an expected result?*
>
> **Response.** Yes, this pattern is expected and stems from both the prior design and clinical reality.
>
> Methodologically, BayesENDS’ event-relational graph (ERG) is **weakly regularized toward Pearson correlations** via a Fisher-z alignment term. This prior encourages, but does not enforce, agreement between ERG edges and conventional correlation-based connectivity. Because of this, some similarity between ERG-based connectivity and Pearson graphs is by design and reflects the prior.
>
> Clinically, frontotemporal dementia (FTD) and Alzheimer’s disease (AD) are known to share overlapping network-level disruptions, particularly in frontal and temporal regions, even though their topographies differ from healthy controls. In our ERG connectivity maps, AD and FTD both show increased long-range frontotemporal disruption relative to HC, and their mutual similarity is stronger than either is to HC. This is consistent with the expectation that both dementias depart from the healthy connectome in related ways.
>
> #### Q2@bzcn
>
> *Another example is with the frequency distributions. Would someone see the shift in central frequency by looking at the EEG trace itself? Is BayesENDS bringing better insights above and beyond such simple spectral analysis for already diagnosed patients?*
>
> **Response.** Raw EEG traces do contain subtle slowing in AD (e.g., longer-period oscillations), and standard power spectral density (PSD) analysis can reveal group-level shifts in alpha/theta bands. However, BayesENDS goes beyond simple PSD in several ways:
>
> 1. **Single-subject frequency latents.** BayesENDS infers **per-subject distributions of latent event frequencies** with credible intervals, rather than just average PSD across epochs. This allows us to quantify, for each subject, how much their dominant oscillatory rates deviate from the healthy distribution.
>
> 2. **Joint modeling of frequencies and connectivity.** The same latent event process that drives the frequency distributions also feeds the ERG connectivity graph. This yields a **joint biomarker**, we can say, for example, that a subject shows alpha-band slowing specifically in frontal channels that also exhibit altered connectivity to temporal regions. Simple PSD does not tie frequency shifts to interaction patterns.
>
> 3. **Task-coupled explainability.** BayesENDS is trained **end-to-end for classification**, so the learned frequency distributions are directly linked to the decision boundary.
>
> #### Q3@bzcn
>
> *“The discussion is terse on interpretability. For example it is not very clear what the graph on EEG connectivity brings for interpretability. See questions below for precise items.”*
>
> **Response.** We appreciate the reviewer’s comment and clarify the role of the connectivity graph in interpretability.
> In BayesENDS, the event-relational graph (ERG) depicts how inferred latent events propagate across electrodes. Edges reflect directed, timing-based influences, so a strong weight from channel A to B indicates that A’s events tend to precede B’s in the latent dynamics. This yields an interpretable map of “drivers’’ and “responders’’ rather than the symmetric co-fluctuations captured by Pearson correlations.
>
> At the group level, cohort-averaged ERGs reveal systematic alterations: AD and FTD subjects show disrupted long-range frontotemporal and posterior interactions relative to healthy controls, consistent with established EEG findings. Comparing graph properties, such as relative frontal and posterior hub strength, links BayesENDS outputs to familiar notions of connectivity degradation in dementia.
>
> At the single-subject level, the ERG offers a compact summary of how an individual’s network departs from healthy structure. Because the ERG directly informs the classifier, edges that deviate most from the control distribution are those that most strongly shape the diagnostic prediction, enabling statements like “this subject shows abnormally strong frontotemporal driving connections’’ rather than relying on opaque latent features.
>
> Finally, a Fisher-z alignment term softly regularizes the ERG toward Pearson connectivity, ensuring that the learned directed graph remains grounded in a familiar EEG measure and can be interpreted alongside standard correlation-based connectivity.

---

> > ### Author Response · Authors · 2025-12-04
> >
> > #### Q4@bzcn
> >
> > *“The recovery of boundary time cross plots is introduced abrubtly and the synthetic data protocol only mentioned in the appendix. This would probably need more context in the main text. Additionally the corresponding Figure 4 is buried in the appendix. It looks like the authors have moved this part between the main text and the appendix and left some information behind.”*
> >
> > **Response.** We agree this was confusing and have reorganized the text.
> >
> > - We now introduce the **toy dataset and boundary-time recovery** explicitly in Sec. 5.1, before referring to Figure 4 and Table 3.
> > - Figure 4 remains in the main paper but is now referenced and described clearly: we explain what the diagonal plots show, why deviation from the identity line indicates boundary-time errors, and how BayesENDS compares to STRODE.
> > - Appendix E.2 contains the full synthetic data generation protocol, and the main text now explicitly points to it instead of assuming the reader will find it.
> >
> > This reorganization aims to make the simulation study and its connection to interpretability more transparent.
> >
> > #### Q5@bzcn
> >
> > *“define IVP and STRODE”*
> >
> > **Response.** Thank you for pointing this out. In the revised manuscript we now define these terms at first use:
> >
> > - **IVP** stands for *initial value problem*, referring to the ODE formulation where the solution trajectory is determined by an initial state and a vector field. In our context, the IVP-based bound approximates the KL between the EPDE-induced event rate and the dLIF prior by bounding the solution of an ODE given an initial condition.
> >
> > - **STRODE** is short for Stochastic boundary ordinary differential equation (Huang et al., 2021).
> >
> >   Huang H, Liu H, Wang H, et al. Strode: Stochastic boundary ordinary differential equation[C]//International Conference on Machine Learning. PMLR, 2021: 4435-4445.

---

### Official Review · Reviewer_yiJh · 2025-11-01

**Soundness:** 2
**Presentation:** 2
**Contribution:** 1
**Rating:** 2
**Confidence:** 3

**Summary:**

This paper introduces BayesENDS for interpretable AD diagnosis from EEG. The model integrates electrophysiological principles into a probabilistic framework that infers latent neural events and a time-varying interaction graph directly from multichannel EEG data. Empirically, the model outperforms CNN, RNN, attention-based, and transformer-based EEG baselines across 2 public AD datasets, and has higher accuracy while producing interpretable physiological biomarkers.

**Strengths:**

1. This paper proposes a new Bayesian neural dynamical system for AD diagnosis using EEG data.
2. The method shows consistent gains over diverse baselines across multiple real-world EEG datasets.

**Weaknesses:**

1. Although the method tries to combine ephys-inspired mechanisms into a model for EEG data, the motivation is not very clear. Since EEG has poor temporal resolution, it may not be appropriate to make such a connection. I would appreciate some clarification of the motivation behind this.

2. The writing is not very clear, and the visualizations can be improved. Many paragraphs are not fully developed (e.g., the related work section).

3. Although the results are promising, the experiments are limited to EEG-based AD datasets. Testing on other neurodegenerative or cognitive tasks would demonstrate that the proposed method is robust.

**Questions:**

1. Given that the model involves multiple variational components, how computationally demanding is it? Could the authors provide a theoretical or empirical runtime comparison with baseline methods?

2. While the inferred latent frequencies and connectivity patterns appear biologically plausible, can the authors validate these findings against ground truth (for example, through simulation) to further support the interpretability claims?

---

> ### Author Response · Authors · 2025-12-04
>
> #### Q1@yiJh
>
> *Given that the model involves multiple variational components, how computationally demanding is it? Could the authors provide a theoretical or empirical runtime comparison with baseline methods?*
>
> **Response.** We appreciate this question. Although BayesENDS introduces several variational components (EPDE, MELP, dLIF prior, ERG), its overall computational footprint remains moderate. Concretely, for a single 2-s EEG window (19 channels, 500 Hz), BayesENDS requires **60.176M FLOPs** and **3.103M parameters**, which is larger than lightweight backbones such as EEGNet and LCADNet, but substantially smaller than more expressive sequence and transformer baselines such as LSTM, ATCNet, ADFormer, and LEAD.
>
> | Model     | FLOPs   | Params   |
> | --------- | ------- | -------- |
> | BayesENDS | 60.176M | 3.103M   |
> | EEGNet    | 9.989M  | 1.232K   |
> | LCADNet   | 7.783M  | 151.302K |
> | LSTM      | 114.82M | 5.5K     |
> | ATCNet    | 328.74M | 4.3K     |
> | ADFormer  | 3.636G  | 5.34M    |
> | LEAD      | 0.766G  | 3.33M    |
>
> #### Q2@yiJh
>
> *While the inferred latent frequencies and connectivity patterns appear biologically plausible, can the authors validate these findings against ground truth (for example, through simulation) to further support the interpretability claims?*
>
> **Response.** Thank you for this suggestion. Our current submission already includes a ground-truth validation of the latent dynamics on a synthetic dataset, but we agree it was not sufficiently highlighted in the main text.
>
> In the toy dataset, we generate sequences from renewal processes with known event rates corresponding to three frequency bands \([5–10], [10–15], [15–20]\) Hz. For this synthetic setting we know the true boundary times and event rates. We then compare BayesENDS with NODE, ODE-RNN, and STRODE.
>
> As shown in **Figure 4** and **Table 3**, BayesENDS accurately recovers the true boundary times and event rates:
>
> - It attains high cosine similarity (CS) between predicted and ground-truth trajectories, **and**
> - It achieves **non-zero IoU** between inferred and true event structures across all bands (e.g., IoU ≈ 0.47 for [5–10] Hz, 0.29 for [10–15] Hz, 0.20 for [15–20] Hz), whereas NODE/ODE-RNN/STRODE have IoU ≈ 0.
>
> This simulation directly confirms that the learned latent frequencies track the underlying oscillatory structure rather than acting as an uninterpretable embedding. In the revision we move a short description of this protocol and a pointer to Figure 4/Table 3 into the main text, and we explicitly state that these results serve as a ground-truth validation of the event-frequency latents.
>
> #### Q3@yiJh
>
> *“Although the method tries to combine ephys-inspired mechanisms into a model for EEG data, the motivation is not very clear. Since EEG has poor temporal resolution, it may not be appropriate to make such a connection. I would appreciate some clarification of the motivation behind this.”*
>
> **Response.** We apologize for the lack of clarity here and also for a possible confusion in terminology. Scalp EEG in fact has **high temporal resolution** (on the order of milliseconds) but **low spatial resolution**. This high temporal precision is precisely what motivates modeling mesoscopic, event-driven dynamics with electrophysiology-inspired priors.
>
> Our motivation is not to interpret the dLIF prior as a literal single-neuron model at the scalp, but as a **phenomenological prior on population-level rate dynamics**. The dLIF prior constrains latent event rates to exhibit leaky accumulation, refractoriness, and oscillatory ranges consistent with known EEG phenomena in AD (e.g., slowing from alpha/beta to theta). These constraints regularize an otherwise flexible neural ODE and encourage latents that align with biologically plausible frequency ranges, as validated on the synthetic data and by the group-level spectral shifts in AD cohorts.
>
> #### Q4@yiJh
>
> *“Although the results are promising, the experiments are limited to EEG-based AD datasets. Testing on other neurodegenerative or cognitive tasks would demonstrate that the proposed method is robust.”*
>
> **Response.** We agree that demonstrating robustness across more tasks would be valuable. Our choice to focus on two AD EEG cohorts was driven by space constraints and by our aim to analyze explainability by any limitation of BayesENDS to AD thoroughly.
>
> A full multi-task evaluation is beyond the current page limits, but we agree it is a natural direction and explicitly mention it as future work.

---

### Author Response · Authors · 2025-12-04

We thank the reviewers for their careful and thoughtful evaluations. We are encouraged that they found our work “novel” (rzUB), our theory “thorough” and “clear,” and our presentation “clear and modular” (P5qn).
Most concerns raised by the reviewers relate to the clarity of the writing, which in turn affected their ratings. We have taken this feedback seriously and have made substantial improvements to the paper's presentation accordingly.

---

### Meta-Review · Area_Chair_2FtM · 2026-01-03

**Summary:**

This paper proposes BayesENDS, a Bayesian electrophysiology-inspired neural dynamical system for Alzheimer’s disease diagnosis from EEG. The model combines an EEGNet-style encoder with latent event dynamics (EPDE/MELP), a differentiable LIF-inspired prior (dLIF), and an event-relational directed graph (ERG) to produce both classification outputs and putatively interpretable biomarkers (latent frequency structure and connectivity). Reviewers agreed the topic is important and several found the modeling approach and theoretical treatment (including the IVP-based bound for the dLIF prior) interesting and potentially novel. However, the initial submission raised major concerns around experimental/reporting clarity, evaluation protocol, reproducibility, and how strongly the interpretability and biophysical grounding claims are supported for scalp EEG. After the authors’ rebuttal, many clarity and reproducibility gaps were improved, but remaining concerns about empirical strength, generality, and the interpretation/validation of biomarkers keep the paper below the ICLR acceptance bar.

**Reviewer Concerns:**

The authors’ rebuttal clears up several of the main points that were confusing in the initial submission. In particular, it makes the evaluation protocol explicit (strict cross-subject splitting with no subject leakage), states the windowing choice (2 s, non-overlapping), and clarifies that the EEGNet front-end is trained jointly end-to-end rather than being pretrained or frozen. The rebuttal also fills in missing experimental details and adds the additional metrics reviewers asked for (e.g., sensitivity, specificity, AUC). On the modeling side, the authors more carefully frame the dLIF component as a coarse-grained, phenomenological prior on latent rate dynamics rather than a literal single-neuron explanation of scalp EEG, and they point to the synthetic experiments as a ground-truth check that the event-rate latents track known structure. Overall, these clarifications make the setup easier to interpret and improve reproducibility.

However, several substantive issues remain. The empirical evidence is still concentrated on two AD EEG cohorts with limited sample sizes, and it is unclear how robust the approach is outside this setting (other cohorts, other neurodegenerative conditions, or other EEG tasks). The interpretability story, while better motivated, is still largely qualitative: the biomarker claims rely heavily on visual plots, and the paper would be stronger with more quantitative group comparisons, subject-level uncertainty, and a clearer connection between specific biomarkers and the model’s decisions. The discussion of related work also needs to be tighter. The authors’ point about protocol differences (cross-subject vs epoch-wise) is reasonable, but the paper would benefit from a more direct, apples-to-apples comparison and clearer reconciliation with prior reported numbers. Finally, concerns about stability remain: reviewers noted large variance in ablation results and asked for stronger statistical support, and while reporting has improved, it is still hard to judge how consistent the gains from each prior are across folds and settings.

In short, the rebuttal improves clarity and reproducibility substantially, but the paper still feels promising rather than fully convincing, especially with respect to breadth of validation and the rigor of the interpretability claims.

**Reviewer Scores:**

With the additional clarifications, it is reasonable to expect that concerns centered on presentation and missing implementation details would be alleviated to some extent. That said, the core issues identified in the reviews are largely unchanged, and most scores would likely stay close to their original values. The most positive reviewer (bzcn, score 6) would probably remain at a marginal-accept level. Reviewers who focused on experimental design and protocol clarity (P5qn, G8qL) would likely appreciate the improved exposition, but would still have open questions about the breadth of evaluation, the strength of the biomarker analysis, and the robustness of the reported gains, making a substantial score increase unlikely. Reviewers who were more skeptical of the connection between EEG and the proposed electrophysiological priors, as well as the limited empirical scope (yiJh, rzUB), would likely acknowledge the improved motivation and clearer protocol, but would still lean toward rejection given the remaining concerns about external validity, interpretability quantification, and positioning relative to prior work.

---

### Decision · Program_Chairs · 2026-01-26

Reject